# StreamSplat: Towards Online Dynamic 3D Reconstruction from Uncalibrated Video Streams

**Zike Wu**[1,2]  **Qi Yan**[1,2]  **Xuanyu Yi**[4]  **Lele Wang**[1]  **Renjie Liao**[1,2,3]
[1]University of British Columbia  [2]Vector Institute for AI
[3]Canada CIFAR AI Chair  [4]Nanyang Technological University
{zikewu, qi.yan, lelewang, rjliao}@ece.ubc.ca, xuanyu001@e.ntu.edu.sg

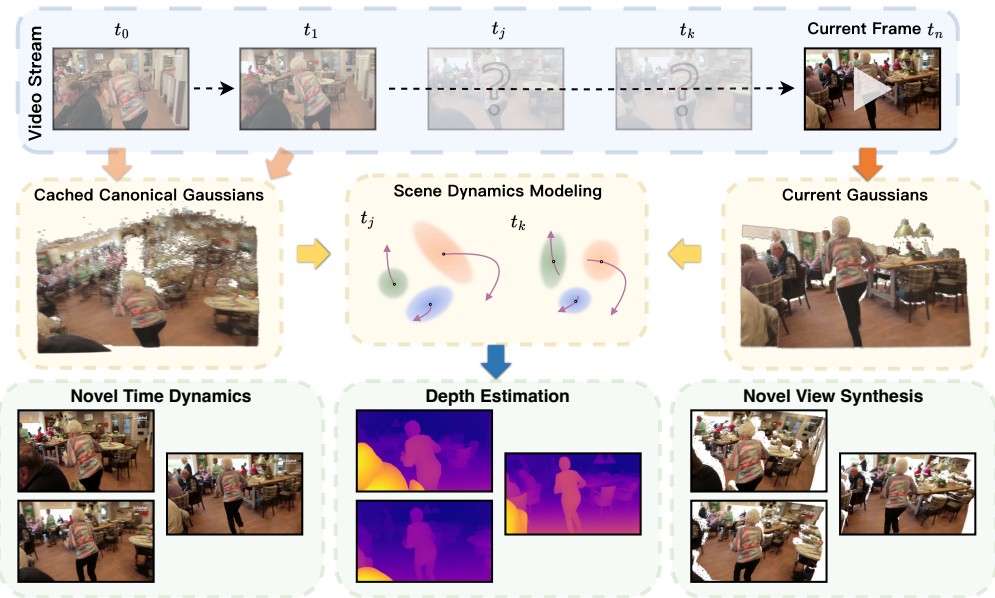

Figure 1: Given an uncalibrated video stream, **StreamSplat** instantly reconstructs a dynamic 3D Gaussian scene in an online manner, enabling continuous-time 3D reconstruction, depth estimation, and novel view synthesis.

## Abstract

Real-time reconstruction of dynamic 3D scenes from uncalibrated video streams demands robust online methods that recover scene dynamics from sparse observations under strict latency and memory constraints. Yet most dynamic reconstruction methods rely on hours of per-scene optimization under full-sequence access, limiting practical deployment. In this work, we introduce **StreamSplat**, a fully feed-forward framework that instantly transforms uncalibrated video streams of arbitrary length into dynamic 3D Gaussian Splatting (3DGS) representations in an online manner. It is achieved via three key technical innovations: 1) a probabilistic sampling mechanism that robustly predicts 3D Gaussians from uncalibrated inputs; 2) a bidirectional deformation field that yields reliable associations across frames and mitigates long-term error accumulation; 3) an adaptive Gaussian fusion operation that propagates persistent Gaussians while handling emerging and vanishing ones. Extensive experiments on standard dynamic and static benchmarks demonstrate that StreamSplat achieves state-of-the-art reconstruction quality and dynamic scene modeling. Uniquely, our method supports the online reconstruction of arbitrarily long video streams with a $1200\times$ speedup over optimization-based methods. Our code and models are available at https://streamsplat3d.github.io/.

## 1 INTRODUCTION

Real-time dynamic 3D reconstruction, *a.k.a.* 4D reconstruction, from video streams is crucial for numerous applications, *e.g.*, robotics (Huang et al., 2023; 2024), augmented/virtual reality (AR/VR) (Carmigniani and Furht, 2011; Cruz-Neira et al., 1992), and autonomous driving (Sun et al., 2020). Specifically, AR/VR systems rely on accurate, continuously updated 3D scenes for immersive experiences, while robots and autonomous vehicles require real-time representations of dynamic environments for safe navigation and responsive interaction.

The prevailing approach to dynamic 3D reconstruction, however, still relies on *offline*, per-scene optimization (Park et al., 2021a;b). Although recent dynamic 3D Gaussian Splatting (3DGS) methods (Stearns et al., 2024; Lei et al., 2025) have reduced processing time from days to hours, they continue to follow a multi-step offline pipeline: (i) camera calibration (Schonberger and Frahm, 2016) and optimization of *static 3D Gaussians* for each key frame; (ii) optimization of a learnable *per-Gaussian deformation field* across the entire sequence to capture intermediate dynamics, with the Gaussian set fixed (Luiten et al., 2024; Duan et al., 2024; Wang et al., 2024a); and (iii) application of *temporal aggregation and fusion* (*e.g.* divide-and-conquer (Stearns et al., 2024), k-NN (Lei et al., 2025), or interpolation (Lee et al., 2024)) to ensure temporal coherence.

Despite recent advances, existing pipelines remain fundamentally *offline*. They require access to the entire video sequence and hours of iterative, per-scene computation, making them impractical for real-world applications under sparse observations and strict latency constraints. This challenge leads to a crucial research question: *Can we match the quality and functionality of offline methods while operating fully online with uncalibrated video streams?*

We answer this question affirmatively by introducing **StreamSplat**, an efficient, scalable, and feed-forward framework for online dynamic 3D reconstruction. Given uncalibrated video streams, StreamSplat directly predicts dynamic 3DGS representations in near real-time. Crucially, it preserves core principles of dynamic 3D reconstruction: (i) maintaining a persistent 3D state via *canonical 3D Gaussians* predicted from observations, (ii) modeling dynamics through a *per-Gaussian deformation field*, and (iii) enforcing temporal coherence through *streaming aggregation and fusion* (Figure 1).

To enable this online reconstruction capability, we propose three technical innovations: 1) *Probabilistic position sampling*: A mechanism that robustly predicts 3D Gaussians from uncalibrated inputs. This approach captures geometric uncertainty and avoids local minima common in feed-forward models (Charatan et al., 2024) (Section 3.1). 2) *Bidirectional deformation field*: A method that estimates both forward and backward motion between the canonical state and the current observation. This yields reliable associations across frames and mitigates long-term error accumulation (Section 3.2). 3) *Adaptive Gaussian fusion*: An operation naturally suited for streaming data that performs soft matching to propagate *persistent* Gaussians while effectively handling *emerging* and *vanishing* ones (Section 3.2). In summary, our key contributions are:

- We introduce **StreamSplat**, an efficient, scalable, feed-forward framework for *online dynamic 3D reconstruction from uncalibrated video streams*.
- We propose three technical innovations: 1) a probabilistic sampling mechanism for 3DGS position prediction, 2) a bidirectional deformation field for robust, efficient dynamic modeling, and 3) an adaptive fusion to maintain a temporally coherent dynamic 3DGS representation.
- We evaluate our framework on both dynamic (DAVIS (Pont-Tuset et al., 2017), YouTube-VOS (Xu et al., 2018)) and static (CO3Dv2 (Reizenstein et al., 2021), RealEstate10K (Zhou et al., 2018)) benchmarks, across *video reconstruction*, *frame interpolation*, and *novel view synthesis*. Our method achieves state-of-the-art performance with a $1200\times$ speedup over optimization-based baselines, while uniquely supporting fully online reconstruction of arbitrarily long streams (Section 4).

## 2 RELATED WORKS

**Dynamic 3D Reconstruction.** Early approaches use implicit MLPs optimized per scene (Ouyang et al., 2024; Chen et al., 2021; Sitzmann et al., 2020; Tancik et al., 2020), or add time-conditioned deformation fields (Pumarola et al., 2021; Park et al., 2021a;b; Li et al., 2022), but still struggle with large, complex motion. Recent methods adopt explicit 3D primitives, particularly dynamic

Gaussian Splattings (Luiten et al., 2024; Li et al., 2024; Duan et al., 2024; Yuan et al., 2025; Sun et al., 2024; Lee et al., 2024; Shi et al., 2025), for efficient, real-time rendering. However, they still rely on calibrated camera poses and extensive per-scene optimization, making them unsuitable for fully uncalibrated and real-time scenarios.

**Feed-Forward Reconstruction.** By directly predicting 3D scene representations via neural networks, feed-forward methods have emerged as promising alternatives to optimization-based approaches. Recent SLAM-based (Matsuki et al., 2024; Yan et al., 2024; Yugay et al., 2023; Zhu et al., 2024) and scene-coordinate-based methods (Wang et al., 2024b; Chen et al., 2025; Jang et al., 2025; Leroy et al., 2024; Xu et al., 2024a; Smart et al., 2024; Yang et al., 2025; Wang et al., 2025a) can estimate camera parameters and 3D structure from uncalibrated inputs but target only static scenes. MonST3R extends to dynamic scene estimations (Zhang et al., 2024a), but typically requires pose/geometry post-optimization, and coordinate-based point clouds remain hard to deform (Luiten et al., 2024; Duan et al., 2024). On the other hand, feed-forward 3DGS methods achieve fast static reconstruction (Wang et al., 2024c; Chen et al., 2024; Zhang et al., 2024b; Shen et al., 2025a; Yi et al., 2024; Xu et al., 2024b; Hong et al., 2024; Zhang et al., 2025; Liu et al., 2024): pixelSplat regresses Gaussians from calibrated pairs (Charatan et al., 2024), NoPoSplat handles uncalibrated pairs (Ye et al., 2024), and StreamGS enables online static streams (Li et al., 2025). However, recent dynamic variants (Liang et al., 2024; Yang et al., 2024a; Shen et al., 2025b) still rely on camera calibration and full-sequence access, thus failing to support uncalibrated online streaming. In contrast, **StreamSplat** operates strictly online on uncalibrated inputs, maintaining temporally consistent dynamic Gaussians via bidirectional deformation and adaptive Gaussian fusion.

## 3 METHOD

In this section, we present **StreamSplat**, a framework designed to instantly transform uncalibrated online video streams into dynamic 3D Gaussian Splatting (3DGS) representations capable of capturing scene dynamics. Figure 2 provides an overview. We first encode the current frame into static 3D Gaussians in a canonical space (Section 3.1), then predict a *bidirectional* deformation field between Gaussians encoded from current frame and propagated from the previous frame, and finally fuse them into a unified dynamic representation (Section 3.2). This representation supports rendering at arbitrary times and viewpoints, thereby effectively recovering scene dynamics (Section 3.3).

### 3.1 PROBABILISTIC 3D GAUSSIAN ENCODING

**Canonical 3D Space.** To support a unified model for in-the-wild videos with unknown and varying intrinsics (e.g., rectilinear/pinhole and fisheye), we adopt a shared orthographic canonical space following Wang et al. (2023); Sun et al. (2024); Shen et al. (2025b) (details in Appendix A). This choice bypasses per-scene camera calibration: camera motion and perspective effects are absorbed into the Gaussian dynamics and handled by our *Dynamic Decoder* (Section 3.2). For a detailed discussion on perspective projection, see Appendix D. This canonical space provides a simplified, yet effective foundation for predicting dynamic 3D representations directly from uncalibrated videos.

**Structured Static 3D Gaussian Encoding.** To overcome the inherent depth ambiguity under orthographic projection (Chen et al., 2016) and unstructured nature of 3DGS (Chung et al., 2024), we incorporate a pretrained depth estimator (Yang et al., 2024b) and predict 3D Gaussian positions in a pixel-aligned manner (Charatan et al., 2024).

Given an input frame $I$, we form an RGB-D image (via a pseudo-depth) and partition it into $8 \times 8$ patches. A Transformer-based *Static Encoder* (Vaswani et al., 2017) produces *3DGS embeddings* $\mathbf{h}$. A lightweight upsampler (Liu et al., 2021; Xu et al., 2024c) then yields per-$2 \times 2$-patch Gaussian tokens $\hat{\mathbf{E}}$. Each token $\hat{\mathbf{E}}_i$ is decoded by linear heads into 3DGS parameters: position offset $\boldsymbol{o}_i \in [-1, 1]^3$, rotation $\mathbf{R}_i$, scale $\mathbf{S}_i$, opacity $\boldsymbol{\alpha}_i$, and color $\boldsymbol{c}_i$. The final 3D position is computed via pixel-aligned prediction: $\boldsymbol{\mu}_i = (u_i + \boldsymbol{o}_{i,0}, v_i + \boldsymbol{o}_{i,1}, g(\boldsymbol{o}_{i,2}))$, where $(u_i, v_i)$ is the pixel coordinate, $\boldsymbol{o}_{i,2}$ denotes the inverse depth (Yang et al., 2024b) for better depth estimation near the camera, and $g(z) = 2/(1+z)$ is the depth mapping. Algorithm 1 includes this procedure and detailed architecture is in Figure 9.

**Probabilistic Position Sampling.** 3D Gaussian Splatting is sensitive to position initialization (Kerbl et al., 2023) and prone to local minima (Charatan et al., 2024), especially in feed-forward models (Charatan et al., 2024; Yi et al., 2024) that make predictions in a single forward pass without

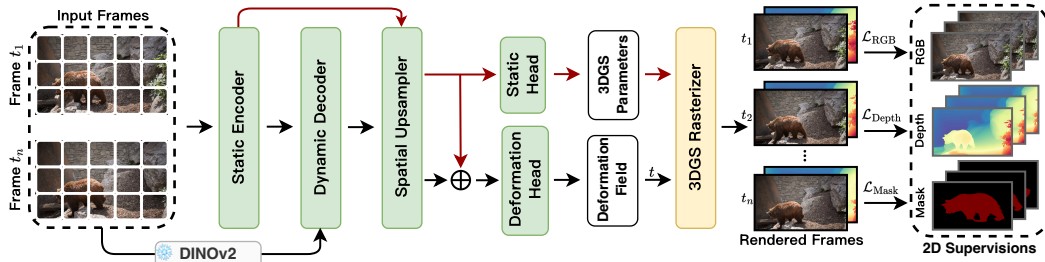

Figure 2: Overview of the **StreamSplat**. Given a pair of frames ($t_1 = 0, t_n = 1$), we first encode them using the *Static Encoder* to produce canonical 3D Gaussians (Section 3.1), and then pass the 3DGS Embeddings to the *Dynamic Decoder* to predict the deformation field (Section 3.2). The predicted dynamic 3D Gaussians can be rendered at arbitrary time $t \in [0, 1]$.

iterative refinement. Inspired by Charatan et al. (2024), we predict a *truncated normal distribution* for each 3D offset $o$ rather than regressing it directly: $o \sim \mathcal{N}_{[-1,1]}(\boldsymbol{\mu}_p, \boldsymbol{\Sigma}_p)$, where $\boldsymbol{\mu}_p$ and $\boldsymbol{\Sigma}_p$ are the predicted mean and covariance. As shown in Section 4.4, this strategy promotes spatial exploration during early training and stabilizes convergence toward optimal positions.

## 3.2 DYNAMIC DEFORMATION PREDICTION

Online dynamic reconstruction involves large non-rigid motions and topological changes (*e.g.*, emerging/vanishing surfaces). We address this with two modules: (i) a *bidirectional deformation field* that provides reliable cross-frame associations and naturally handles newly appearing and disappearing content; and (ii) an *adaptive Gaussian fusion* mechanism that implicitly merges and propagates persistent Gaussians, maintaining long-term temporal coherence. Per-Gaussian deformation is parameterized by a 3D velocity $\mathbf{v} \in [-1, 1]^3$ and an opacity coefficient $\boldsymbol{\gamma}$ that controls visibility over time. Likewise, $\mathbf{v}$ is sampled by the same probabilistic position sampling mechanism (Algorithm 1).

**Bidirectional Deformation Field.** When a new frame $I_t$ arrives, optimization-based methods (Duan et al., 2024; Lei et al., 2025) typically instantiate new Gaussians and refine them iteratively. However, this strategy is hard to cast into a feed-forward model: learning a *variable* number of Gaussians typically requires model selection (Akaike, 1998) or nonparametric priors like Dirichlet processes (Ferguson, 1973)), which complicate the end-to-end training.

Instead, we propose to jointly model forward and backward motion between consecutive frames: the *forward* field deforms previous-frame Gaussians $\mathcal{G}_{t-1}$ to current time $t$, and the *backward* field deforms current-frame Gaussians $\mathcal{G}_t$ back to previous time $t - 1$. This symmetric formulation yields robust cross-frame associations and naturally handles *emerging* and *vanishing* Gaussians in a unified manner, thereby simplifying prediction and supervision for end-to-end training and online inference.

**Adaptive Gaussian Fusion via Soft Matching.** Directly combining new Gaussians often leads to spatial overlap and redundancy (Li et al., 2025). Traditional optimization-based methods resolve this by rigid one-to-one matching and iterative fusion (Lee et al., 2024; Stearns et al., 2024), which are computationally expensive and hard to keep spatial structures, potentially resulting in long-term accumulation of errors.

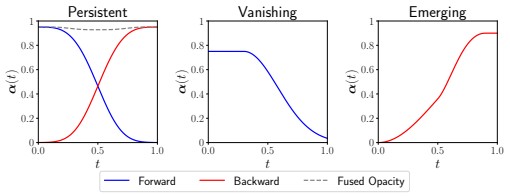

Figure 3: Our *opacity deformation* jointly models persistent, emerging, and vanishing Gaussians.

Inspired by defining the *life-cycle* of Gaussians (Zhao et al., 2024), we propose an *adaptive Gaussian fusion* mechanism based on opacity deformation. Each two-frame interval is normalized to $t \in [0, 1]$, with $t = 0$ and $t = 1$ representing the previous and current frames, respectively. Each Gaussian persists across two consecutive frames, with a time-dependent opacity deformation:

$$\boldsymbol{\alpha}(t) = \boldsymbol{\alpha} \cdot \frac{\sigma\left(-\boldsymbol{\gamma}_0\left(|t - t_0| - \boldsymbol{\gamma}_1\right)\right)}{\sigma\left(\boldsymbol{\gamma}_0 \cdot \boldsymbol{\gamma}_1\right)}, \tag{1}$$

where $\sigma(\cdot)$ denotes the sigmoid function, $\boldsymbol{\alpha}$ denotes initial opacity, $t_0 \in \{0, 1\}$ denotes the Gaussian's creation frame, $\boldsymbol{\gamma}_0 \in \mathbb{R}^+$ and $\boldsymbol{\gamma}_1 \in [0, 1]$ controls transition rate and fade-out window. Modulating overlap via time-dependent opacity *implicitly* fuses the forward and backward Gaussians: the recon-

| **Algorithm 1:** Dynamic 3DGS Prediction | **Algorithm 2:** Online Inference |
|---|---|
| **Input** : Embedding $\mathbf{h}_{t_0}$, $\mathbf{d}_{t_0}$, frame time $t_0$ 
 **Output** : Dynamic 3DGS $\mathcal{G}(t)$ | **Input** : Frame $I_k$, canonical Gaussians $\tilde{\mathcal{G}}(t)$, stored $\mathbf{h}_{k-1}$, camera pose $\pi$ 
 **Output** : Updated $\tilde{\mathcal{G}}(t)$, stored $\mathbf{h}_k$, rendered frames $\hat{I}_{t_{k-1} \to t_k}$ |

**Algorithm 1:** Dynamic 3DGS Prediction

**Input** : Embedding $\mathbf{h}_{t_0}$, $\mathbf{d}_{t_0}$, frame time $t_0$
**Output** : Dynamic 3DGS $\mathcal{G}(t)$

```
// Static
```
$(\boldsymbol{\mu}_p, \boldsymbol{\Sigma}_p, \mathbf{S}, \mathbf{R}, \boldsymbol{\alpha}_0, \boldsymbol{c}) \leftarrow \text{HEAD}(\mathbf{h}_{t_0});$
$[\boldsymbol{o}_0, \boldsymbol{o}_1, \boldsymbol{o}_2] \sim \mathcal{N}_{[-1,1]}(\boldsymbol{\mu}_p, \boldsymbol{\Sigma}_p);$
$\boldsymbol{\mu}_0 \leftarrow [u + \boldsymbol{o}_0, v + \boldsymbol{o}_1, g(\boldsymbol{o}_2)];$

```
// Dynamic
```
$(\Delta\boldsymbol{\mu}_p, \Delta\boldsymbol{\Sigma}_p, [\boldsymbol{\gamma}_0, \boldsymbol{\gamma}_1]) \leftarrow \text{HEAD}(\mathbf{h}_{t_0}, \mathbf{d}_{t_0});$
$\mathbf{v} \sim \mathcal{N}_{[-1,1]}(\Delta\boldsymbol{\mu}_p, \Delta\boldsymbol{\Sigma}_p);$

```
// Deformation
```
$\boldsymbol{\mu}(t) \leftarrow \boldsymbol{\mu}_0 + \mathbf{v} \cdot (t - t_0);$
$\boldsymbol{\alpha}(t) \leftarrow \boldsymbol{\alpha}_0 \cdot \dfrac{\sigma\big(-\boldsymbol{\gamma}_0\,(|t - t_0| - \boldsymbol{\gamma}_1)\big)}{\sigma(\boldsymbol{\gamma}_0 \cdot \boldsymbol{\gamma}_1)};$
$\mathcal{G}(t) \leftarrow \{(\boldsymbol{\mu}(t), \boldsymbol{\alpha}(t), \mathbf{S}, \mathbf{R}, \boldsymbol{c})\};$

**return** $\mathcal{G}(t)$

**Algorithm 2:** Online Inference

**Input** : Frame $I_k$, canonical Gaussians $\tilde{\mathcal{G}}(t)$, stored $\mathbf{h}_{k-1}$, camera pose $\pi$
**Output** : Updated $\tilde{\mathcal{G}}(t)$, stored $\mathbf{h}_k$, rendered frames $\hat{I}_{t_{k-1} \to t_k}$

```
// Prediction
```
$\mathbf{h}_k \leftarrow \text{ENCODER}(I_k);$
$(\mathbf{d}_{k-1}^+, \mathbf{d}_k^-) \leftarrow \text{DECODER}(\mathbf{h}_{k-1}, \mathbf{h}_k);$
$\mathcal{G}_{k-1}^+(t) \leftarrow \text{PRED}(\mathbf{h}_{k-1}, \mathbf{d}_{k-1}^+, t_{k-1});$
$\mathcal{G}_k^-(t) \leftarrow \text{PRED}(\mathbf{h}_k, \mathbf{d}_k^-, t_k);$ `// Alg. 1`

```
// Update Deformation & Fusion
```
$\tilde{\mathcal{G}}(t) \leftarrow \text{UPDATE}\big(\tilde{\mathcal{G}}(t), \mathcal{G}_{k-1}^+(t)\big) \cup \mathcal{G}_k^-(t);$
$\hat{I}_{t_{k-1} \to t_k} \leftarrow \text{RASTERIZE}(\tilde{\mathcal{G}}(t), \pi);$
$\tilde{\mathcal{G}}(t) \leftarrow \{g \in \tilde{\mathcal{G}}(t) \mid \boldsymbol{\alpha}_g(t_k) > 0\};$

**return** $\tilde{\mathcal{G}}(t), \mathbf{h}_k, \hat{I}_{t_{k-1} \to t_k}$

struction loss induces *soft match* that propagate **persistent** Gaussians while handling **emerging** and **vanishing** ones (Figure 3), thereby maintaining temporal coherence without hard assignments or iterative fusion. In Section 4.2, Figure 4 and Figure 15 further shows that, despite viewpoint/scale changes, motion, occlusions, and blur, our adaptive Gaussian fusion still preserves long-term temporal coherence with persistent Gaussians.

**Dynamic Deformation Decoding.** Given two consecutive frames $I_{t-1}$ and $I_t$, we use the frozen static encoder to obtain *3DGS embeddings* $\mathbf{h}_{t-1}$ and $\mathbf{h}_t$. A Transformer-based *Dynamic Decoder* conditioned on DINOv2 features (Oquab et al., 2023) produces *Deformation Embeddings*. The same upsampler yields per-$2\times2$-patch deformation tokens $\hat{\mathbf{d}}$, which are concatenated with their paired static tokens $\hat{\mathbf{E}}$ and decoded by a small MLP into per-Gaussian *velocity* and *opacity coefficients*. These define forward/backward deformation fields that move and fade Gaussians over time, recovering continuous-time scene evolution. Algorithm 2 includes the procedure; see Appendix B for details.

### 3.3 TRAINING AND INFERENCE

**Robust Stage-wise Training.** To address the difficulty of jointly optimizing static 3D Gaussian encoding and dynamic deformation prediction, we adopt a two-stage training protocol.

**Stage 1: Static 3DGS Encoder Training.** The static encoder aims to reconstruct static 3DGS from a single input frame $I_t$ and pseudo-depth $D_t$. It produces 3DGS primitives that are used to render RGB $\hat{I}_t$ and depth $\hat{D}_t$. The training loss combines photometric and depth supervision:

$$\mathcal{L}_{\text{static}} = \mathcal{L}_{\text{recon}}(\hat{I}_t, I_t) + \lambda_{\text{depth}}\mathcal{L}_{\text{depth}}(\hat{D}_t, D_t), \qquad (2)$$

where $\mathcal{L}_{\text{recon}}$ is the sum of $L_2$ loss (RGB space) and LPIPS loss (Zhang et al., 2018), and $\mathcal{L}_{\text{depth}}$ is a scale- and shift-invariant depth loss (Ranftl et al., 2020):

$$\mathcal{L}_{\text{depth}} = \mathbb{E}\|\tau(\hat{D}_t) - \tau(D_t)\|, \quad \text{where} \quad \tau(\mathbf{x}) = \frac{\mathbf{x} - \text{median}(\mathbf{x})}{\mathbb{E}\|\mathbf{x} - \text{median}(\mathbf{x})\|}. \qquad (3)$$

To reduce the impact of noisy pseudo-depth, we introduce an adaptive decay factor into the depth loss weight. Specifically, we define the effective weight as $\hat{\lambda}_{\text{depth}} = \lambda_{\text{depth}} \cdot \sigma(-\|\tau(\hat{D}_t) - \tau(D_t)\|/w)$, where $\lambda_{\text{depth}}$ is a fixed hyperparameter, and $w$ controls the sensitivity of the sigmoid-based decay. We use $\hat{\lambda}_{\text{depth}}$ instead of $\lambda_{\text{depth}}$ to reduce unreliable supervision and improve robustness.

**Stage 2: Dynamic Deformation Decoder Training.** With the encoder frozen, the dynamic decoder learns to predict the bidirectional deformation fields. Given two randomly sampled frames $I_{t_1}$ and $I_{t_2}$, we first encode them to obtain static 3DGS $\mathcal{G}_{t_1}$ and $\mathcal{G}_{t_2}$. Then we use the dynamic decoder to predict the deformation fields that transform $\mathcal{G}_{t_1} \to I_{t_2}$ and $\mathcal{G}_{t_2} \to I_{t_1}$, followed by adaptive fusion. The training objective is to reconstruct $I_t$ for all intermediate time $t \in [t_1, t_2]$:

$$\mathcal{L}_{\text{dynamic}} = \mathbb{E}_t\left[\mathcal{L}_{\text{recon}}(\hat{I}_t, I_t) + \lambda_{\text{depth}}\mathcal{L}_{\text{depth}}(\hat{D}_t, D_t) + \lambda_{\text{mask}}\mathcal{L}_{\text{mask}}(\hat{I}_t \odot M_t, I_t \odot M_t)\right], \qquad (4)$$

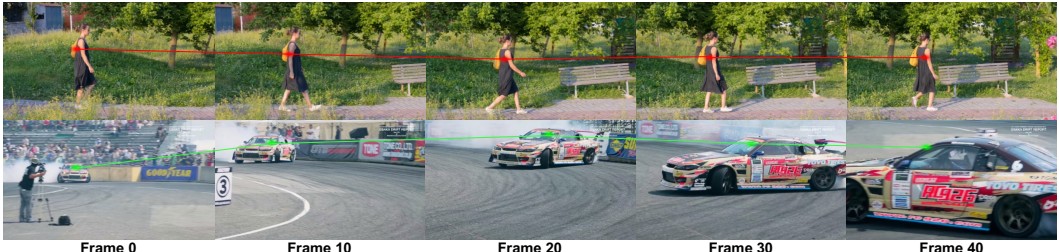

Figure 4: **Persistent Gaussians across frames.** Red/green-marked Gaussians from initial frame are propagated across frames, showing that adaptive Gaussian fusion preserves long-term temporal consistency under viewpoint and motion changes. Videos are available on the project website.

where $\mathcal{L}_{\text{mask}}$ is an auxiliary reconstruction loss that encourages the model to focus on moving foreground regions using a binary segmentation mask $M_t$ from datasets (Pont-Tuset et al., 2017; Xu et al., 2018).

**Online Inference Pipeline.** Algorithm 2 summarizes the online inference. We maintain a canonical Gaussian set $\tilde{\mathcal{G}}(t)$ and store the previous embedding $\mathbf{h}_{k-1}$. For each incoming frame $I_k$, we predict pseudo-depth $D_k$, encode $\mathbf{h}_k$, and use the *Dynamic Decoder* with Algorithm 1 on $(\mathbf{h}_{k-1}, \mathbf{h}_k)$ to produce two Gaussian sets: $\mathcal{G}_{k-1}^+(t)$ (previous Gaussians deformed forward) and $\mathcal{G}_k^-(t)$ (current Gaussians deformed backward). We then (i) UPDATE $\tilde{\mathcal{G}}(t)$ with $\mathcal{G}_{k-1}^+(t)$, setting the active deformation of matched 3DGS primitives to the new forward field, (ii) aggregate and fuse $\mathcal{G}_k^-(t)$, (iii) render frames $\hat{I}_{t_{k-1} \to t_k}$ from given viewpoints, and (iv) prune Gaussians whose opacity at $t_k$ decays to zero. In the end, we store $\mathbf{h}_k$ for the next step.

## 4 EXPERIMENTS

### 4.1 EXPERIMENTAL SETTINGS

**Training Datasets.** We pre-train **StreamSplat** on a combination of real-world datasets, including static scenes from CO3Dv2 (Reizenstein et al., 2021) and RealEstate10K (RE10K) (Zhou et al., 2018), and dynamic video datasets DAVIS (Pont-Tuset et al., 2017) and YouTube-VOS (Xu et al., 2018). CO3Dv2 and RE10K are treated as pure video datasets without using any pre-calibrated camera information. For dynamic scenes, we utilize object segmentation masks from DAVIS and YouTube-VOS to supervise motion-aware components, and no mask supervision is applied to CO3Dv2 and RE10K. We follow the official train/validation splits and train only on the training sets. The same pre-trained model is evaluated across both static and dynamic benchmarks.

**Implementation Details.** **StreamSplat** is trained on 8 NVIDIA A100 GPUs for approximately 3 days. We use FlashAttention-2 (Dao, 2023), gradient checkpointing (Chen et al., 2016), and mixed-precision training with BF16 for better efficiency. Input frames are resized to $512 \times 288$ to preserve aspect ratio for pixel-aligned 3DGS prediction. We apply image-level augmentations following EDM (Karras et al., 2022) and optimize using AdamW (Loshchilov and Hutter, 2017) with gradient clipping set to 1.0. For **Stage 1**, we use a batch size of 128, a peak learning rate of $5 \times 10^{-4}$ with 20K linear warm-up iterations, and weight decay of 0.05. For **Stage 2**, we use a batch size of 256 (with gradient accumulation), a peak learning rate of $1 \times 10^{-4}$ with 100K linear warm-up iterations, and weight decay of 0.05. Additional training details are in Appendix B.

**Evaluation Settings.** We follow prior works (Charatan et al., 2024; Jain et al., 2024) and report peak signal-to-noise ratio (PSNR), structural similarity index (SSIM) (Wang et al., 2004), and LPIPS (Zhang et al., 2018), all evaluated at a resolution of $256 \times 256$ for fair comparison (Charatan et al., 2024; Jain et al., 2024). For static scene reconstruction, we follow (Charatan et al., 2024), using two randomly sampled input views with at least $60\%$ overlap and five target views sampled between them. For dynamic scene reconstruction, we evaluate on sparse, uncalibrated videos subsampled at 5- and 8-frame intervals (Jain et al., 2024) and compute metrics on all non-input frames. We also conducted a zero-shot evaluation on DyCheck (Gao et al., 2022) and NVIDIA Dynamic Scene (Yoon et al., 2020) in Appendix C. Videos are available on the project website.

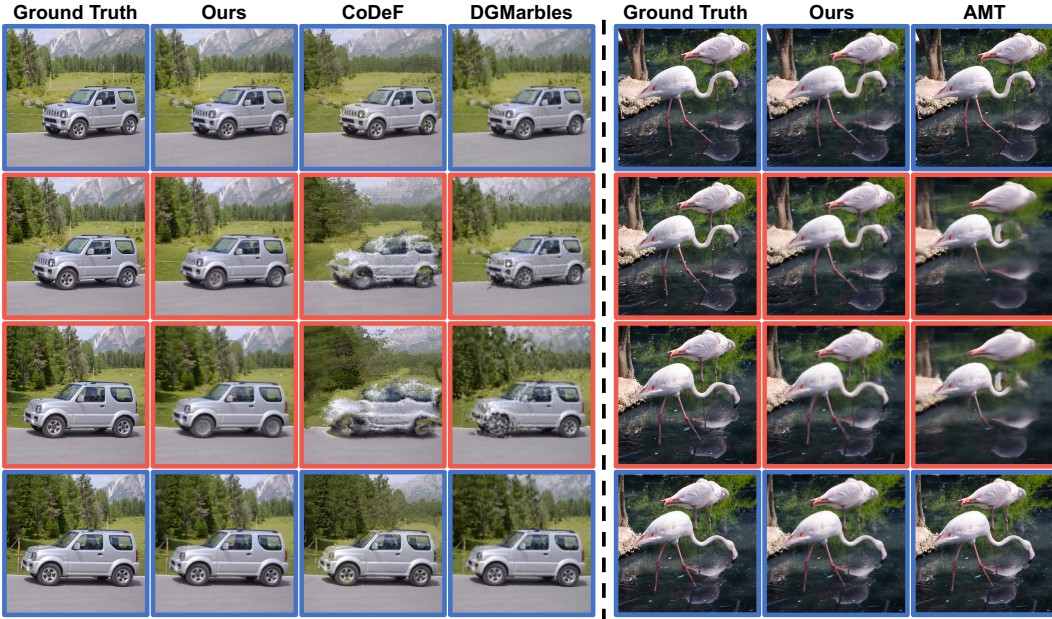

Figure 5: **Qualitative comparison on DAVIS. Blue box:** given frames; **Red box:** novel frames. **StreamSplat** produces high-fidelity and temporal coherent results across both (a) 5-frame and (b) 8-frame interval tasks.

Table 1: **Quantitative results on DAVIS.** [†] denotes results reported in the original papers.

| Method | Scene Rep. | Key Frames | | | Middle-4 Frames | | | Time |
|---|---|---|---|---|---|---|---|---|
| | | PSNR$\uparrow$ | SSIM$\uparrow$ | LPIPS$\downarrow$ | PSNR$\uparrow$ | SSIM$\uparrow$ | LPIPS$\downarrow$ | |
| Omnimotion[†] (Wang et al., 2023) | NeRF | 24.11 | 0.714 | 0.371 | – | – | – | > 8 hrs |
| RoDynRF[†] (Liu et al., 2023) | NeRF | 24.79 | 0.723 | 0.394 | – | – | – | > 24 hrs |
| CoDeF (Ouyang et al., 2024) | NeRF | 31.49 | 0.939 | 0.088 | 19.40 | 0.498 | 0.400 | ~10 mins |
| MonST3R (Zhang et al., 2024a) | Points | **42.33** | 0.980 | **0.012** | – | – | – | ~30 s |
| 4DGS[†] (Duan et al., 2024) | 3DGS | 18.12 | 0.573 | 0.513 | – | – | – | ~40 mins |
| Splater a Video[†] (Sun et al., 2024) | 3DGS | 28.63 | 0.837 | 0.228 | – | – | – | ~30 mins |
| DGMarbles (Stearns et al., 2024) | 3DGS | 28.38 | 0.879 | 0.172 | 21.33 | 0.619 | 0.313 | ~30 mins |
| **StreamSplat (Ours)** | 3DGS | 37.83 | **0.982** | 0.016 | **23.66** | **0.684** | **0.193** | ~1.48 s |

## 4.2 DYNAMIC SCENE RECONSTRUCTION

**Video Reconstruction.** We evaluate **StreamSplat** on the DAVIS benchmark and compare it with state-of-the-art methods, and report the main results in Table 1. Notably, **StreamSplat** is the only method capable of near real-time dynamic 3D reconstruction, with a runtime of only **0.049s per-frame** and a 1200× speedup over optimization-based baselines.

For key-frame reconstruction task, **StreamSplat** achieves performance competitive with the state-of-the-art scene-coordinate-based method MonST3R (Zhang et al., 2024a), which represents

Table 2: 8-frame interval interpolation results.

| Method | Type | PSNR$\uparrow$ | SSIM$\uparrow$ | LPIPS$\downarrow$ |
|---|---|---|---|---|
| AMT | | 21.09 | 0.544 | 0.254 |
| RIFE | | 20.48 | 0.511 | 0.258 |
| FILM | 2D | 20.71 | 0.528 | 0.270 |
| LDMVFI | | 19.98 | 0.479 | 0.276 |
| VIDIM | | 19.62 | 0.470 | 0.257 |
| CoDeF | | 20.34 | 0.520 | 0.365 |
| DGMarbles | 4D | 19.83 | 0.548 | 0.353 |
| **Ours** | | **22.10** | **0.613** | **0.234** |

the dynamic scenes as sequences of *static* 3D point clouds. However, MonST3R requires extensive post-optimization and is **limited** to key-frame reconstruction. In contrast, **StreamSplat** operates in near real time and explicitly models the scene dynamics, enabling reconstruction of intermediate frames across substantial temporal gaps.

We further evaluate dynamic modeling performance under 5-frame and 8-frame interval settings, comparing against optimization-based 3D reconstruction methods (CoDeF (Ouyang et al., 2024), DGMarbles (Stearns et al., 2024)) and 2D video-interpolation methods (AMT (Li et al., 2023),

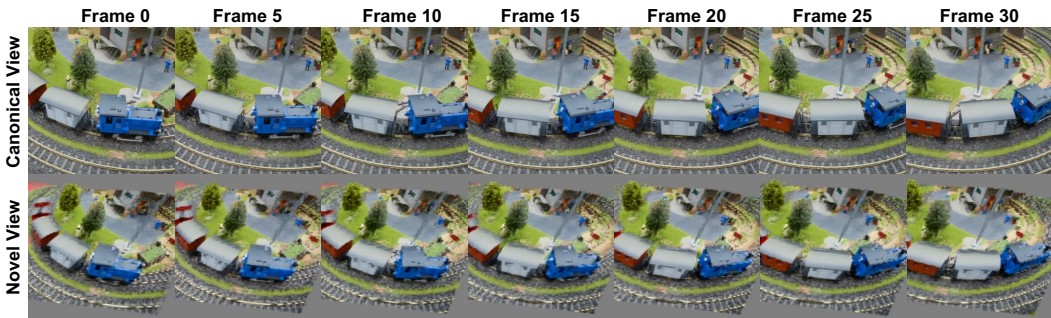

Figure 6: **Visualization of reconstructed scene from canonical and novel views.** Our method captures consistent 3D scene over time, enabling faithful reconstruction at arbitrary time and viewpoints.

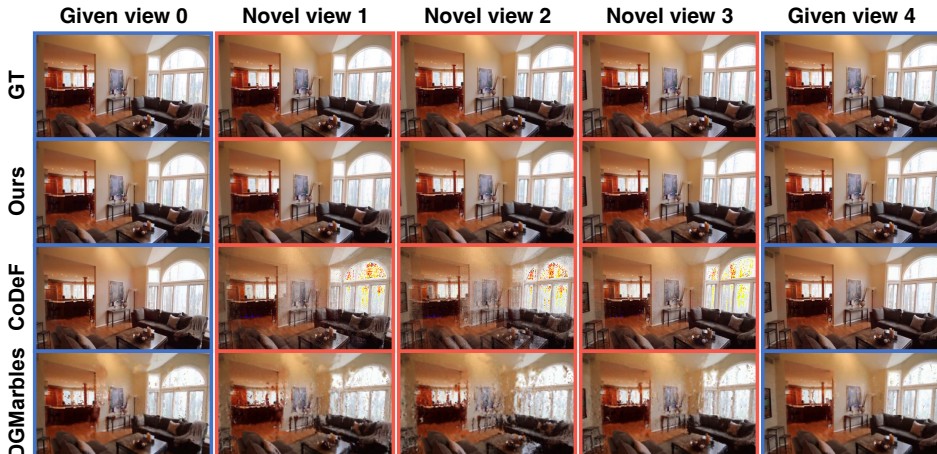

Figure 7: **Qualitative results on RE10K.** Blue box: given frames; Red box: novel frames. **Stream-Splat** produces consistent reconstructions, while others exhibit color/geometry distortions.

RIFE (Huang et al., 2022), FILM (Reda et al., 2022), VIDIM (Jain et al., 2024)). As shown in Table 2, Figures 5 and 12, **StreamSplat** consistently outperforms all baselines, including video-interpolation methods which lack explicit 3D modeling. This highlights the effectiveness of our approach in modeling temporally coherent dynamic scenes. Moreover, as illustrated in Figures 5 and 12, **StreamSplat** maintains high visual fidelity even under challenging scenarios such as large camera motion and reflective surfaces, where other methods often fail.

In addition, we demonstrate the robustness of **StreamSplat** in long-range dynamic modeling and novel view synthesis. As shown in Figure 6, our method maintains coherent 3D structure and appearance across large spatio-temporal distances, from both canonical and novel viewpoints.

**Temporal Coherence and Persistency.** We evaluate the temporal coherence of our modeled representation by color-coding random Gaussians in the initial frame and propagating their identities through the fused Gaussians over time (Figures 4 and 15). Despite viewpoint/scale changes, motion, occlusions, and blur, the highlighted Gaussians remain stable, indicating that our adaptive fusion maintains long-term consistency without explicit matching or iterative fusion.

### 4.3 STATIC SCENE RECONSTRUCTION

We evaluate **StreamSplat** on RE10K against recent *dynamic* reconstruction methods; results for static methods and CO3Dv2 are deferred to Appendix C. For pose-free dynamic pipelines, novel views are rendered using relative timestamps. Quantitative and qualitative results are presented in Table 3 and 5, Figures 7, 11 and 10.

Table 3: **RE10K results.** PSNR$^\uparrow$/LPIPS$^\downarrow$ for 2 given views and 5 novel views.

| Method | Given View | Novel View | Average |
|---|---|---|---|
| CoDeF | 35.13 / 0.09 | 20.51 / 0.40 | 21.77 / 0.37 |
| DGMarbles | 27.48 / 0.23 | 23.40 / 0.33 | 23.73 / 0.32 |
| **StreamSplat** | **41.60 / 0.01** | **24.68 /0.167** | **29.51 / 0.12** |

**StreamSplat** significantly outperforms all static baselines on the given-view reconstruction, and

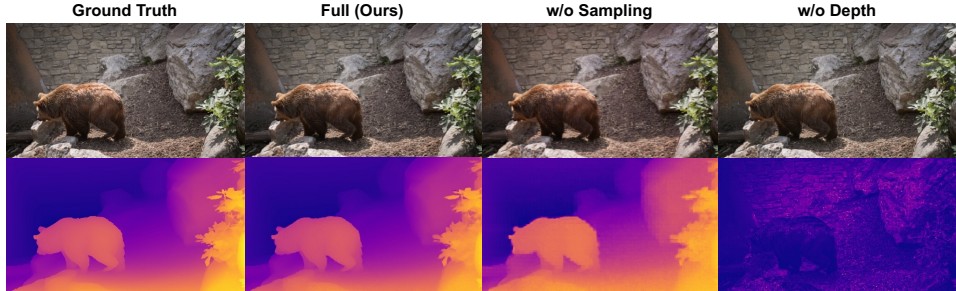

Figure 8: **Ablation.** w/o sampling: deterministic position prediction; w/o depth: no depth supervision.

consistently outperforms all dynamic baselines in every evaluation setting, with an average gain of 5.78dB in PSNR. Qualitatively, in Figure 7, **StreamSplat** produces more detailed and consistent 3D reconstructions across diverse scenes, whereas other dynamic reconstruction methods often exhibit distortions in both color and geometry.

## 4.4 ABLATION STUDIES

We conduct ablation studies to evaluate the contribution of each key component in StreamSplat. Quantitative and qualitative results are presented in Table 4 and Figures 8 and 13.

**Ablation on probabilistic position sampling.** We evaluate the impact of probabilistic position sampling for 3DGS by comparing it with a deterministic counterpart. As shown in Table 4, probabilistic sampling yields a substantial improvement

Table 4: Component-wise ablations on key and intermediate frames.

| Variants | Frame | PSNR$^\uparrow$ | SSIM$^\uparrow$ | LPIPS$^\downarrow$ |
|---|---|---|---|---|
| w/o Sampling | | 31.47 | 0.946 | 0.073 |
| w/o Depth | *Key* | 36.68 | 0.975 | 0.039 |
| **Full (Ours)** | | **37.83** | **0.982** | **0.016** |
| w/o Bi-Deform. | *Mid.* | 18.89 | 0.582 | 0.492 |
| **Full (Ours)** | | **23.66** | **0.684** | **0.193** |

(6.36dB) in PSNR for key-frame reconstruction. Qualitatively, the deterministic variant is prone to local minima, particularly along the depth axis, resulting in blurry and inaccurate reconstructions. This aligns with findings from prior work (Kerbl et al., 2023; Charatan et al., 2024) and highlights the importance of probabilistic position prediction in feed-forward 3DGS models.

**Ablation on pseudo depth supervision.** We evaluate the impact of removing pseudo depth supervision on key-frame reconstruction. Table 4 shows that removing depth supervision leads to only a minor drop in reconstruction quality. However, as shown in Figure 8, the model without depth supervision fails to capture accurate spatial structure. The learned depth becomes entangled with RGB values, resulting in distorted 3D reconstructions.

**Ablation on bidirectional deformation field.** We evaluate the effectiveness of the bidirectional deformation field by comparing it with a conventional deformation field (Li et al., 2024) on middle-frame reconstruction. As shown in Table 4 and Figure 13, the bidirectional variant significantly improves reconstruction quality. The baseline struggles to preserve pixel-aligned structures, leading to noticeable error accumulation over longer sequences.

## 5 CONCLUSION

In this paper, we introduced **StreamSplat**, a fully feed-forward framework for instant, online dynamic 3D reconstruction from uncalibrated video streams. By incorporating a probabilistic position sampling strategy and a bidirectional deformation field with adaptive Gaussian fusion, our **StreamSplat** effectively addresses key challenges in online dynamic reconstruction, allowing to produce accurate dynamic 3D Gaussian Splatting (3DGS) representations from arbitrarily long video streams. These representations faithfully capture scene dynamics and support rendering at arbitrary time and viewpoints. Extensive experiments on both static and dynamic benchmarks validate the superior performance of **StreamSplat** in terms of reconstruction quality and dynamic scene modeling. In future, we plan to explore its potential in video generation and autonomous driving.

ACKNOWLEDGEMENTS

This work was supported, in part, by the NSERC DG Grant (No. RGPIN-2022-04636, No. RGPIN-2019-05448), the Vector Institute for AI, the Canada CIFAR AI Chair program, a Google Gift Fund, and a NVIDIA Academic Grant. Resources used in preparing this research were provided, in part, by the Province of Ontario, the Government of Canada through the Digital Research Alliance of Canada `www.alliancecan.ca`, and companies sponsoring the Vector Institute `www.vectorinstitute.ai/#partners`, and Advanced Research Computing at the University of British Columbia. Additional resource was provided by John R. Evans Leaders Fund CFI grant. ZW and QY are supported by UBC Four Year Doctoral Fellowships. We thank Qiuhong Shen and Qingshan Xu for constructive discussions and helpful comments.

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

# Appendices

The Appendix is organized as follows:

- **Appendix** A: **Background.** A brief overview of 3D Gaussian Splatting (3DGS), including the orthographic projection implementation used in our method.
- **Appendix** B: **Implementation Details.** Model configurations, hyper-parameters, details on 3DGS parameterization, including the implementation of our probabilistic position sampling, and details on model framework.
- **Appendix** C: **Additional Experimental Results.** Additional qualitative and quantitative results on static scene reconstruction, novel view synthesis, ablation studies, and video reconstruction.
- **Appendix** D: **Discussions.** A detailed discussion of the rationale and trade-offs behind key design choices.
- **Appendix** E: **Limitations.** A discussion of current limitations and future directions.

## A  BACKGROUND

**3D Gaussian Splatting.** 3D Gaussian Splatting (3DGS) (Kerbl et al., 2023) represents a static scene using a collection of 3D Gaussians. Each Gaussian $\mathcal{G}$ is defined by its mean (position) $\boldsymbol{\mu}$, covariance matrix $\boldsymbol{\Sigma}$, opacity $\boldsymbol{\alpha}$, and color represented by spherical harmonics (SH) coefficients $\boldsymbol{c}$. The final opacity of a 3D Gaussian at a given point $\mathbf{x}$ is computed as:

$$\boldsymbol{\alpha}(\mathbf{x}) = \boldsymbol{\alpha} \cdot \exp\left(-\frac{1}{2}(\mathbf{x} - \boldsymbol{\mu})^T \boldsymbol{\Sigma}^{-1}(\mathbf{x} - \boldsymbol{\mu})\right). \tag{5}$$

Normally, the covariance matrix $\boldsymbol{\Sigma}$ is decomposed into a diagonal scaling matrix $\mathbf{S}$ and a rotation quaternion $\mathbf{R}$ for differentiable optimization:

$$\boldsymbol{\Sigma} = \mathbf{R}\mathbf{S}\mathbf{S}^T\mathbf{R}^T. \tag{6}$$

Given a set of $N$ 3DGS $\mathcal{G} = \{G_i\}_{i=1}^N$, the rendering process involves splatting them onto the 2D image plane and then blending their colors based on their opacity and depth.

The 3D Gaussians are projected using the approximate transformation (Zwicker et al., 2001):

$$\boldsymbol{\Sigma}' = \mathbf{J}\mathbf{W}\boldsymbol{\Sigma}\mathbf{W}^T\mathbf{J}^T, \tag{7}$$

where $\mathbf{J}$ is the Jacobian of the *perspective projection* function defined as:

$$(u, v) = \left(f_x \cdot x/z + c_x, f_y \cdot y/z + c_y\right),$$
$$\mathbf{J} = \frac{\partial(u, v)}{\partial(x, y, z)} = \begin{pmatrix} f_x/z & 0 & -f_x \cdot x/z^2 \\ 0 & f_y/z & -f_y \cdot y/z^2 \end{pmatrix}. \tag{8}$$

In case of *orthographic projection*, the projection is simplified to:

$$(u, v) = \left(f_x \cdot x + c_x, f_y \cdot y + c_y\right),$$
$$\mathbf{J} = \frac{\partial(u, v)}{\partial(x, y, z)} = \begin{pmatrix} f_x & 0 & 0 \\ 0 & f_y & 0 \end{pmatrix}. \tag{9}$$

The pixel color $C$ is obtained by alpha-blending the projected 2D Gaussians sorted by depth:

$$C = \sum_{i=1}^N c_i \boldsymbol{\alpha}_i \prod_{j=1}^{i-1}(1 - \boldsymbol{\alpha}_j) \tag{10}$$

where $\boldsymbol{\alpha}_i$ is the opacity of the $i$-th projected Gaussian obtained by Eq. 5.

**Dynamic 3D Gaussian Splatting.** Dynamic 3D Gaussian Splatting extends the original 3DGS with a deformation field to model the motion of the Gaussians over time (Luiten et al., 2024). Typically, the deformation fields $\mathcal{D}(t) = \{\boldsymbol{\mu}(t), \mathbf{R}(t), \boldsymbol{\alpha}(t)\}$ are used to update the static canonical Gaussian with various of approaches (Li et al., 2024; Duan et al., 2024; Sun et al., 2024; Lee et al., 2024; Shi et al., 2025). For example, given a static position $\mu$ and a deformation field $\mu(t)$, the deformed position can be obtained as: $\hat{\mu}_t = \mu + \mu(t)$.

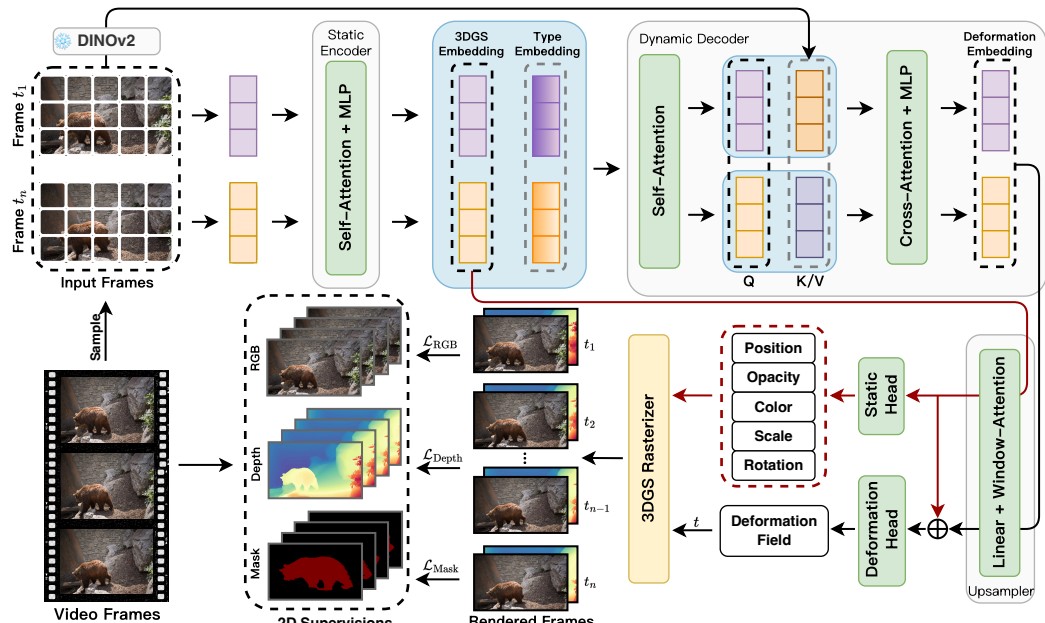

Figure 9: **StreamSplat** framework. Given a pair of frames, we first encode them using the *Static Encoder* to produce canonical 3D Gaussians (Section 3.1), and then pass the 3DGS Embeddings to the *Dynamic Decoder* to predict the deformation field (Section 3.2). The resulting dynamic 3D Gaussians can be rendered at arbitrary time to produce RGB images and depth maps.

## B    IMPLEMENTATION DETAILS

**Model Configurations.** As shown in Table 9, **StreamSplat** consists of an image tokenizer, a static encoder, a dynamic decoder, and an upsampler. The image tokenizer takes RGBD inputs at a resolution of $288 \times 512$ and uses a patch size of 8 to produce 768-dimensional tokens. Both the static encoder and the dynamic decoder contain 10 transformer layers with an embedding dimension of 768 and 12 attention heads. The dynamic decoder uses 0.1 stochastic drop path to prevent overfitting on image features. An upsampler with 2 layers of window attention with 2304 token length increases the token length to $16\times$ and reduces the embedding dimension from 768 to 192. The model uses a linear head for static prediction and a two-layer MLP for deformation prediction. We apply loss balancing weights of 1.0 for MSE, 0.05 for both LPIPS and depth, and 3.0 for the mask loss.

**3DGS Parameterization.** 3DGS parameters are obtained as summarized in Algorithm 1. A linear head projects each Gaussian token $\hat{\mathbf{E}}$ and, through parameter-specific activations, yields the static tuple $(\boldsymbol{\mu}_0, \mathbf{S}, \mathbf{R}, \boldsymbol{\alpha}_0, \boldsymbol{c})$. A 2-layer MLP, conditioned on the concatenation $\hat{\mathbf{E}} \oplus \hat{\mathbf{d}}$, predicts the velocity $\mathbf{v}$ and opacity coefficients $\boldsymbol{\gamma}$, from which the final time-dependent values are computed by integrating the current time $t$. Note, we adopt our proposed *Probabilistic Position Sampling* for all position-related parameters, including position offset $\boldsymbol{o}$ and velocity $\mathbf{v}$, to improve robustness and avoid spatial local minima. Spatial scaling factors $(f_x, f_y) = (256, 144)$, which equals to patch numbers.

**Dynamic Deformation Decoding.** As shown in Figure 9, given two consecutive frames from a randomly sampled time interval, $I_{t_1}$ and $I_{t_2}$, we first use the frozen static encoder to extract their *3DGS Embeddings* $\mathbf{h}_{t_1}$ and $\mathbf{h}_{t_2}$. We also extract DINOv2 features (Oquab et al., 2023) $\mathbf{f}_{t_1}$ and $\mathbf{f}_{t_2}$ from $I_{t_1}$ and $I_{t_2}$, respectively. To distinguish the embeddings, we add learnable *Type Embeddings* (Devlin et al., 2019; Chen et al., 2020) $\mathbf{T}_{src}, \mathbf{T}_{tgt} \in \mathbb{R}^D$ to $\mathbf{h}_{t_1}$ and $\mathbf{h}_{t_2}$, yielding $\hat{\mathbf{h}}_{t_1}$ and $\hat{\mathbf{h}}_{t_2}$. These are processed by the decoder as follows:

$$[\hat{\mathbf{h}}_{t_1}, \hat{\mathbf{h}}_{t_2}] = \text{Self-Attn}([\hat{\mathbf{h}}_{t_1}, \hat{\mathbf{h}}_{t_2}]),$$
$$\hat{\mathbf{h}}_{t_1} = \text{Cross-Attn}(\hat{\mathbf{h}}_{t_1}, \mathbf{f}_{t_2}), \quad \hat{\mathbf{h}}_{t_2} = \text{Cross-Attn}(\hat{\mathbf{h}}_{t_2}, \mathbf{f}_{t_1}) \tag{11}$$
$$[\hat{\mathbf{h}}_{t_1}, \hat{\mathbf{h}}_{t_2}] = \text{FFN}([\hat{\mathbf{h}}_{t_1}, \hat{\mathbf{h}}_{t_2}]),$$

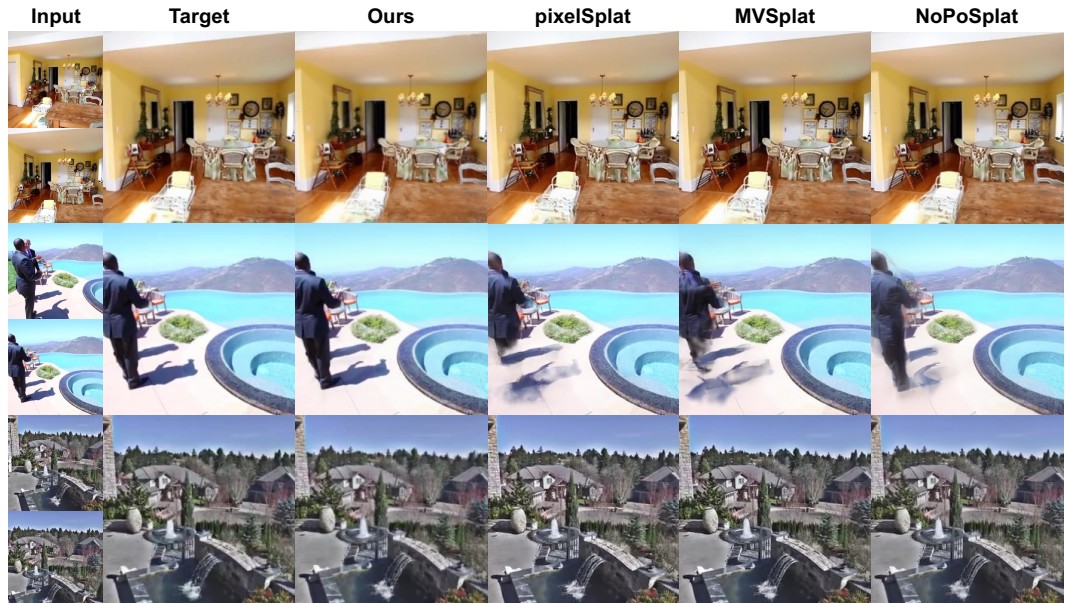

Figure 10: **Qualitative results on RE10K.** StreamSplat produces detailed and consistent reconstructions across diverse scenes.

Table 5: **Quantitative results on RE10K.** We report results for 2 given views and 5 novel views.

| Method | Rep. Type | Given View | | | Novel View | | | Average | | |
|---|---|---|---|---|---|---|---|---|---|---|
| | | PSNR↑ | SSIM↑ | LPIPS↓ | PSNR↑ | SSIM↑ | LPIPS↓ | PSNR↑ | SSIM↑ | LPIPS↓ |
| pixelSplat (Charatan et al., 2024) | | 30.70 | 0.952 | 0.055 | 28.31 | 0.905 | 0.097 | 28.99 | 0.918 | 0.085 |
| MVSplat (Chen et al., 2024) | *Stat.* | 31.48 | 0.962 | 0.046 | 28.48 | 0.909 | **0.091** | 29.34 | **0.924** | **0.078** |
| NoPoSplat (Ye et al., 2024) | | 29.50 | 0.939 | 0.069 | **28.65** | **0.913** | 0.096 | 28.90 | 0.920 | 0.089 |
| CoDeF (Ouyang et al., 2024) | | 35.13 | 0.943 | 0.091 | 20.51 | 0.591 | 0.402 | 21.77 | 0.625 | 0.374 |
| DGMarbles (Stearns et al., 2024) | *Dyn.* | 27.48 | 0.867 | 0.232 | 23.40 | 0.727 | 0.333 | 23.73 | 0.738 | 0.325 |
| **StreamSplat (Ours)** | | **41.60** | **0.992** | **0.010** | 24.68 | 0.777 | 0.167 | **29.51** | 0.839 | 0.122 |

where $[\cdot]$ denotes concatenation. After passing through a few decoder blocks, we obtain the *Deformation Embeddings* $[\mathbf{d}_{t_1}, \mathbf{d}_{t_2}] \in \mathbb{R}^{2N \times D}$, which are then upsampled via the same upsampler to produce deformation tokens $\hat{\mathbf{d}} \in \mathbb{R}^{32N \times D/4}$. Each deformation token $\hat{\mathbf{d}}_j$ is concatenated with the corresponding static Gaussian token from the same frame $\hat{\mathbf{E}}_j$ to form a joint token $\hat{\mathbf{E}}_j \oplus \hat{\mathbf{d}}_j$, which is passed through a 2-layer MLP head to predict the deformation field, including velocity $\mathbf{v}_j$ and opacity coefficient $\boldsymbol{\gamma}_j$. These deformation fields allow Gaussians to move and fade over time, enabling computation of dynamic 3D Gaussians at arbitrary times for continuous scene reconstruction.

## C  ADDITIONAL EXPERIMENTAL RESULTS

Due to the limited space in the main manuscript, we provide additional experimental results in this section, including qualitative and quantitative comparisons with other methods on static scene reconstruction (Figure 10 and 11, Table 5), qualitative comparisons on ablation studies of our bidirectional deformation field (Figure 13), more qualitative results on video reconstruction on CO3Dv2 (Reizenstein et al., 2021), DAVIS (Pont-Tuset et al., 2017), and Youtube-VOS (Xu et al., 2018) (Figure 12), and zero-shot evaluation on DyCheck (Gao et al., 2022) and NVIDIA Dynamic Scene (Yoon et al., 2020) benchmarks (Figure 14, Tables 6 and 7). Please refer to the project website for video results.

**Static Scene Reconstruction** We evaluate **StreamSplat** on the RE10K benchmark and compare it with recent static and dynamic reconstruction methods. For dynamic methods without camera pose input, novel views are rendered using relative timestamps. Quantitative and qualitative results are presented in Table 5 and Figures 7, 10, and 11. **StreamSplat** significantly outperforms all static baselines on the given-view reconstruction task. However, on novel-view reconstruction,

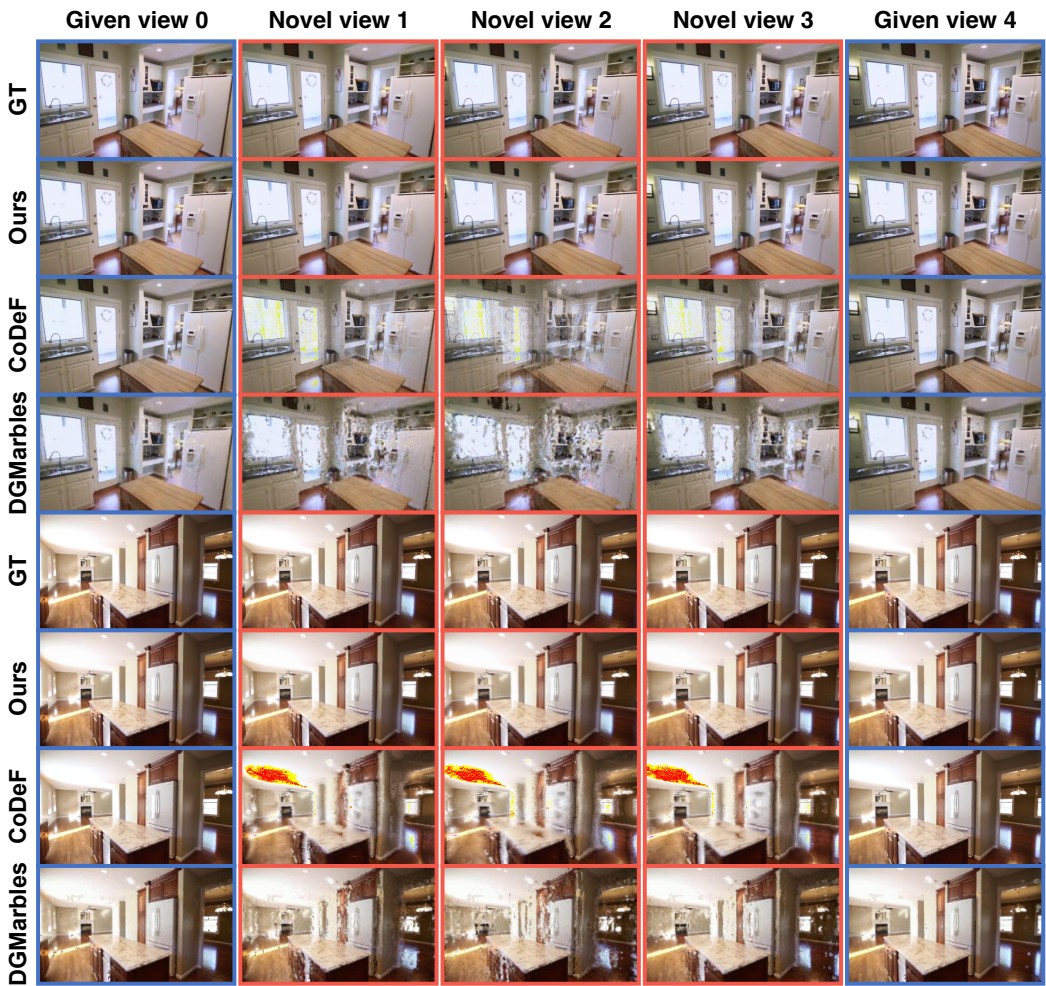

Figure 11: **Qualitative results on RE10K. Blue box:** given frames; **Red box:** novel frames. Compared to other dynamic-scene reconstruction methods, **StreamSplat** produces more detailed and consistent 3D reconstructions across diverse scenes, whereas other methods often exhibit distortions in both color and geometry.

Table 6: **DyCheck results.** We report PSNR$^\uparrow$/LPIPS$^\downarrow$ on novel view synthesis.

| Method | Extr. | Intr. | Apple | Block | Spin | Windmill | Space-out | Teddy | Wheel | Average | Time |
|---|---|---|---|---|---|---|---|---|---|---|---|
| 4DGS (w/ cam)[†] | ✓ | ✓ | 15.41 / .45 | 11.28 / .63 | 14.41 / .34 | 15.60 / .30 | 14.60 / .37 | 12.36 / .47 | 11.79 / .44 | 13.64 / .43 | >6h |
| DGMarbles (w/ cam)[†] | ✓ | ✓ | 17.70 / .49 | 17.42 / .38 | 18.88 / .43 | 17.04 / .39 | 15.94 / .44 | 13.95 / .44 | 16.14 / .35 | 16.72 / .42 | >5h |
| 4DGS (w/o pose)[†] | ✗ | ✓ | 14.44 / .72 | 12.30 / .71 | 12.77 / .70 | 14.46 / .79 | 14.93 / .64 | 11.86 / .73 | 10.99 / .80 | 13.11 / .73 | >6h |
| DGMarbles (w/o pose)[†] | ✗ | ✓ | 16.50 / .50 | 16.11 / .36 | 17.51 / .42 | 16.19 / .45 | 15.97 / .44 | 13.68 / .44 | 14.58 / .39 | 15.79 / .43 | >5h |
| DGMarbles (w/o cam) | ✗ | ✗ | 9.91 / .85 | 10.21 / .81 | 10.64 / .74 | 10.32 / .71 | 10.82 / .71 | 8.33 / .84 | 8.10 / .76 | 9.76 / .77 | >5h |
| **Ours (w/o cam)** | ✗ | ✗ | **12.75 / .74** | **12.06 / .71** | **12.89 / .68** | **13.46 / .67** | **13.95 / .67** | **11.23 / .71** | **10.23 / .69** | **12.37 / .69** | 14s |

Table 7: **NVIDIA Dynamic Scenes results.** We report PSNR$^\uparrow$/LPIPS$^\downarrow$ on novel view synthesis.

| Method | Extr. | Intr. | Scene Info. | Balloon1 | Balloon2 | Jumping | Playground | Skating | Truch | Umbrella | Average |
|---|---|---|---|---|---|---|---|---|---|---|---|
| 4DGS (w/ cam)[†] | GT | GT | ✓ | 14.11 / .40 | 18.56 / .24 | 17.32 / .33 | 13.51 / .34 | 19.41 / .22 | 21.25 / .17 | 19.00 / .34 | 17.59 / .29 |
| DGMarbles (w/ cam)[†] | GT | GT | ✓ | 23.65 / .07 | 21.60 / .14 | 19.61 / .18 | 16.21 / .24 | 24.24 / .09 | 27.18 / .06 | 23.76 / .12 | 22.32 / .13 |
| DGMarbles (w/ CUT3R) | GT | CUT3R | ✓ | 12.66 / .65 | 13.50 / .60 | 14.90 / .53 | 8.12 / .62 | 17.30 / .49 | 15.38 / .55 | 15.50 / .70 | 13.91 / .59 |
| **Ours (w/o cam)** | ✗ | ✗ | ✗ | **14.94 / .48** | **15.77 / .50** | **16.65 / .37** | **12.75 / .58** | **19.08 / .47** | **18.97 / .37** | **15.97 / .60** | **16.30 / .48** |

static methods still lead, although **StreamSplat** achieves the best performance among all dynamic approaches.

According to our qualitative comparison in Figures 10 and 11, we attribute this to the absence of accurate camera poses input in our model. Our **StreamSplat** assumes smooth camera motion between

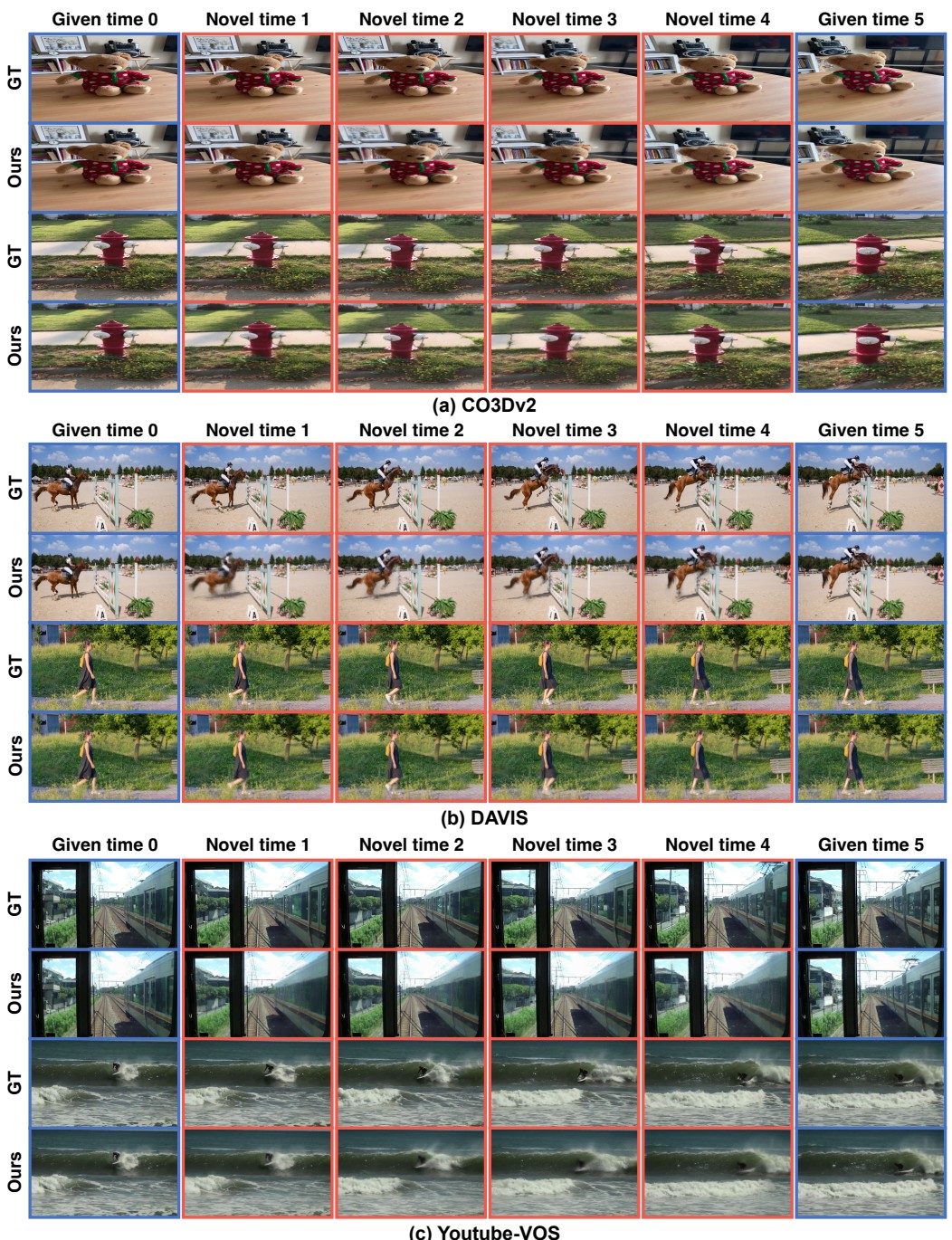

Figure 12: **Qualitative results with 8-frame interval on different datasets.** **Blue box:** given frames; **Red box:** novel frames.

two input views and relies solely on relative timestamps for novel view synthesis, which can cause slight misalignments. Furthermore, unlike static models that assume a fixed scene, **StreamSplat** is designed to model scene dynamics, which may introduce motion artifacts that hinder performance under static benchmarks. Despite these challenges, our **StreamSplat** achieves competitive average performance across all test views and consistently surpasses all dynamic baselines in every evaluation setting. Notably, **StreamSplat** is the only scalable approach among these that requires neither per-scene optimization (Ouyang et al., 2024; Stearns et al., 2024) nor per-dataset training (Charatan et al., 2024; Chen et al., 2024; Ye et al., 2024).

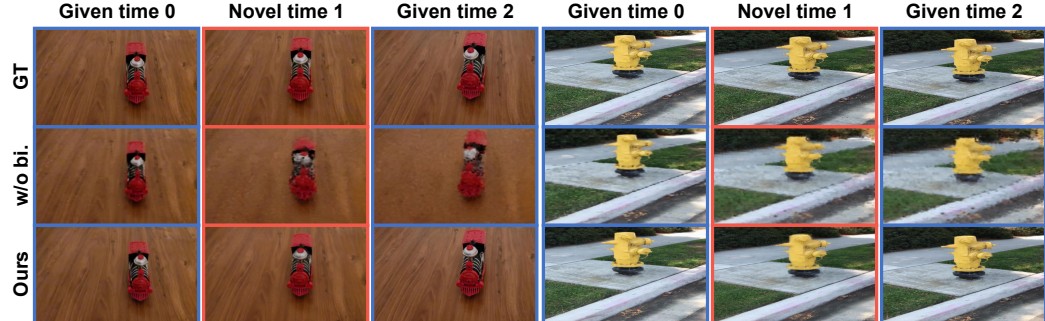

Figure 13: **Ablation.** w/o bi.: without bidirectional deformation field. **Blue box:** given frames; **Red box:** novel frames.

Table 8: Runtime breakdown of **StreamSplat** on a single A100 GPU.

| Module | Time (ms) | Percentage (%) |
|---|---|---|
| Encoder | 18.8 | 38.4 |
| Decoder | 18.4 | 37.6 |
| Rasterization | 4.0 | 8.1 |
| Pre-/Post-processing | 7.8 | 15.9 |
| **Total** | 49 | 100.0 |

**Novel view synthesis on DyCheck benchmark.** We have conducted a **zero-shot** evaluation on 7 scenes from the DyCheck iPhone dataset. To ensure fairness under uncalibrated input, we follow these settings:

- No extrinsics: Fix camera pose and absorb motion into 3D Gaussians.

- No intrinsics: Use a fixed camera intrinsic matrix for all methods. Specifically, for DGMarbles (w/o cam), we use GT principal point and apply only a small perturbation to the focal length to mimic no intrinsics scenarios.

- Evaluation: Compute relative camera pose w.r.t. the fixed pose along DyCheck evaluation trajectories.

The experimental results are provided in Table 6 and Figure 14. From the results, we observe that **StreamSplat (w/o cam)** significantly outperforms DGMarbles (w/o cam) and matches the performance of 4DGS (w/o pose) with GT intrinsic, while being over $1200\times$ faster.

**Novel view synthesis on NVIDIA Dynamic Scene benchmark.** We also conducted a **zero-shot** evaluation on 7 scenes from NVIDIA Dynamic Scene. To ensure fairness under uncalibrated input, we evaluated DGMarbles using intrinsics estimated by CUT3R (Wang et al., 2025b), while providing it with GT poses and scene metadata (scale, near/far) to ensure valid execution. The results are provided in Table 7 and Figure 14. StreamSplat (w/o cam) significantly outperforms DGMarbles (w/ CUT3R) and approaches the performance of 4DGS (w/ cam), which use both GT intrinsic and extrinsic.

**Highly dynamic scenes with topology changes.** Figure 16 illustrates some representative DAVIS sequences with strong non-rigid motion and frequent topology changes, where we visualize our reconstructions across the sequence. Across dancing and acrobatic motions with large limb articulation and self-occlusion, a pedestrian entering and exiting the field of view while passing behind thin structures, and a group of people moving around and mutually occluding each other, **StreamSplat** preserves sharp appearance and consistent geometry over time. Despite objects appearing, disappearing, and undergoing rapid pose changes, the reconstructed frames remain temporally coherent, indicating that the bidirectional deformation and adaptive Gaussian fusion effectively handle highly dynamic, topology-changing real-world scenes.

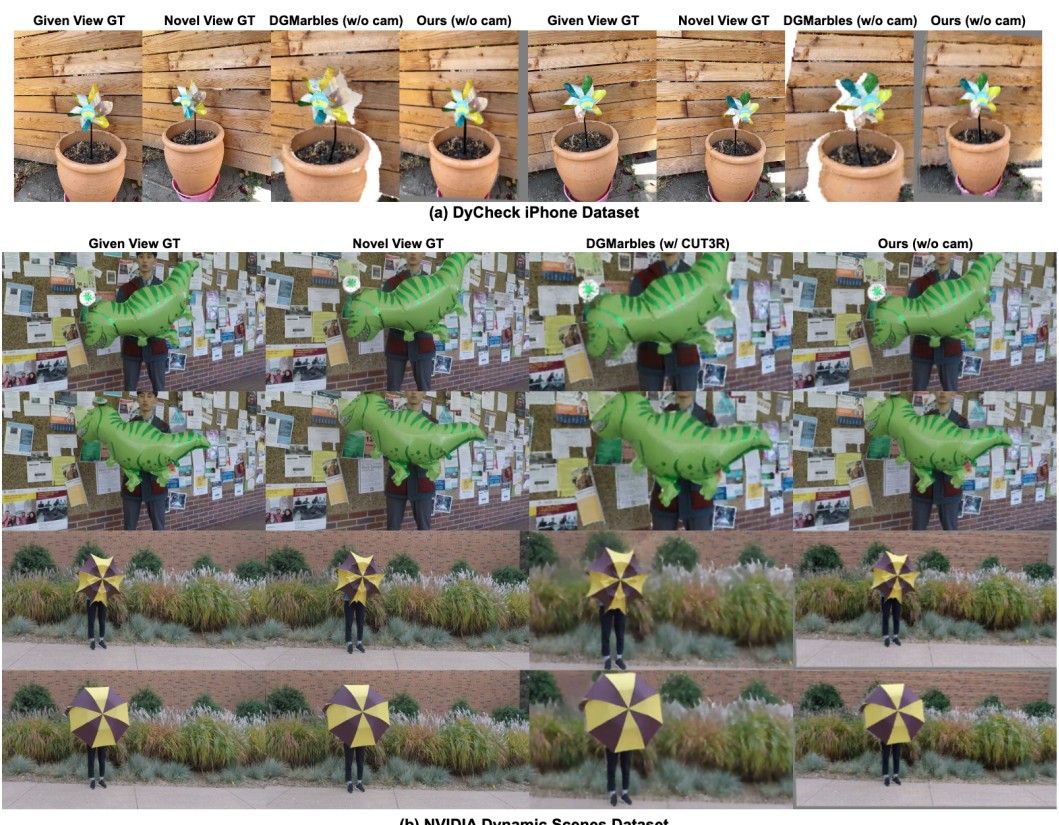

Figure 14: Novel view synthesis results on DyCheck iPhone and NVIDIA Dynamic Scenes.

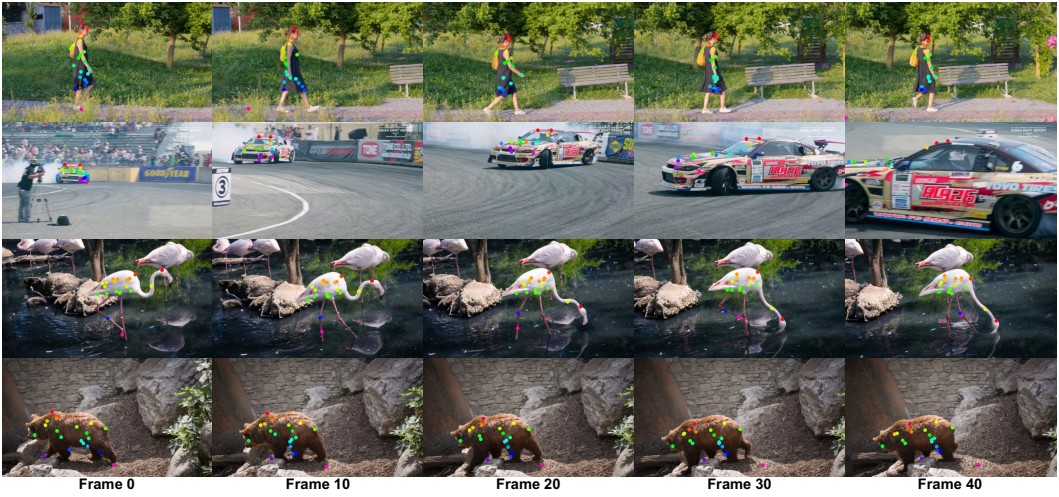

Figure 15: Multiple persistent Gaussians tracking across frames.

**Runtime breakdown.** We report a detailed runtime breakdown of **StreamSplat** in Table 8. On a single NVIDIA A100 GPU, the end-to-end runtime is 0.049s per frame at our default evaluation resolution. The runtime breakdown confirms the practical feasibility of near real-time online reconstruction and suggests that future speedups will primarily come from further optimizing the encoder/decoder components.

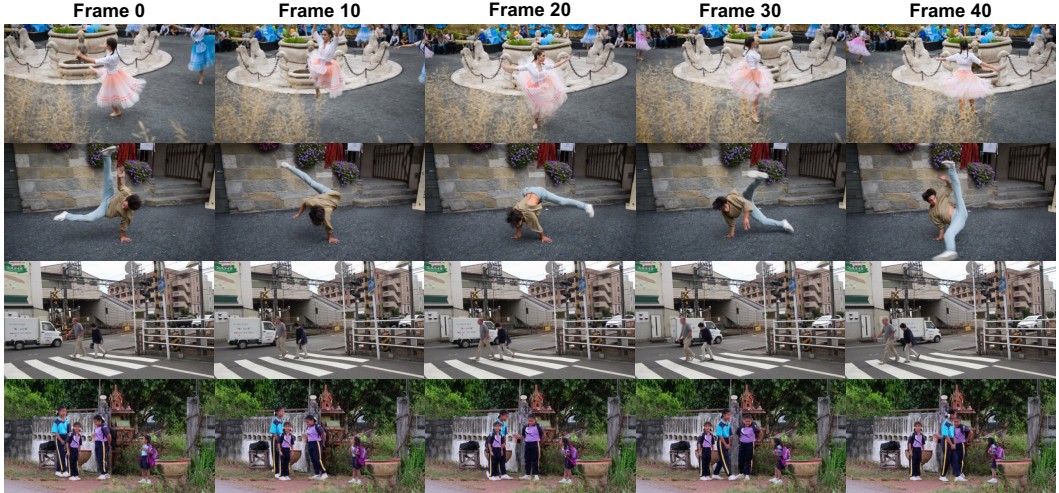

Figure 16: Qualitative results on DAVIS with frequent surface/topology changes and multi-object occlusion.

## D DISCUSSIONS

Our work presents **StreamSplat**, a fully feed-forward framework for online dynamic 3D reconstruction from uncalibrated video streams. Through extensive experiments and analysis, we have demonstrated its effectiveness while also identifying important design choices and trade-offs that need further discussion.

**Orthographic Projection vs. Perspective Projection.** We adopt orthographic projection as a practical choice for handling uncalibrated videos with diverse camera characteristics. This design enables **StreamSplat** to process videos with unknown intrinsics—ranging from standard to wide-angle and fisheye lenses—within a single model. The orthographic formulation decouples geometric reconstruction from camera parameters, absorbing camera motion into the learned Gaussian dynamics.

Our DyCheck experiments validate this choice: DGMarbles (w/o cam) achieves only 10.61 PSNR compared to our 13.46 PSNR on *Windmill*, demonstrating that orthographic projection doesn't compromise accuracy when intergrate with proper deformations. While foreshortening effects occur at close range, our dynamic decoder can deform Gaussians to mimic this perspective effects, as evidenced by strong performance across RE10K and DAVIS benchmarks (Section 4). This robustness to varying camera conditions makes orthographic projection particularly suitable for real-world deployment where calibration is impractical.

In contrast, 3DGS reconstruction is highly sensitive to the assumed camera model: errors in perspective intrinsics (e.g., focal length, principal point) induce scale/shape distortions, for which we are not aware of a simple, proven remedy in online settings. Empirically, even perturbing the intrinsics of DGMarbles (w/o cam) by 10% causes a significant degradation in performance.

**Opacity Deformation.** The bidirectional deformation field employs time-dependent opacity to handle dynamic scene content naturally. This mechanism handles appearing and disappearing objects without explicit tracking and merging, which is crucial for online processing where scene content evolves continuously. Our ablation studies in Figure 8 reveal this: constant opacity cannot accommodate emerging or vanishing content/surfaces, resulting in ghosting and blurring effects.

**Bounded Velocity.** Although naturally velocity can be unbounded, we constrain it within $[-1, 1]^3$ in our implementation. This choice is motivated by two considerations: 1) a velocity magnitude > 1 implies no spatial overlap between consecutive frames, which is essentially impossible in a reconstruction setting. Specifically, magnitude = 1 already means a pixel moves from one image boundary to the opposite boundary within a single frame interval, which is already a very extreme case with large motion. 2) Inferring motion for non-overlapping frames shifts the task toward video/4D generation, which is outside our scope. Thus, there is no need to allow velocity beyond magnitude 1.

**Reconstruction vs. Generation.** We would like to emphasize that **StreamSplat** performs reconstruction, not generation, distinguished by three characteristics:(1) It produces *observation-constrained representations* that minimize reconstruction error while maintaining temporal consistency through Adaptive Gaussian Fusion. (2) The deformation process is deterministic, and novel time/view result from geometric transformations of a fixed set of Gaussians, not probabilistic generating new Gaussians. (3) Gaussians persist across frames via soft matching in canonical space, with opacity deformation handling visibility changes rather than creating new content. Testing with large frame intervals (5-8 frames) challenges temporal coherence more rigorously than dense sampling. Our strong performance under these conditions validates the reconstruction quality, demonstrating that **StreamSplat** maintains scene structure even with significant temporal gaps, which is a capability essential for practical deployment where frame rates may vary.

## E    LIMITATIONS

While **StreamSplat** achieves strong performance, certain design choices introduce some limitations. First, the framework relies on pseudo-depth maps predicted by an external monocular estimator, which may introduce noise–particularly around fine-scale geometry and depth discontinuities. To mitigate this, we incorporate an adaptive decay weighting scheme during training to downweight unreliable depth supervision. Nonetheless, scaling the training dataset to enable internal depth refinement remains a promising direction to reduce reliance on external priors. Second, the bidirectional deformation field is trained over a two-frame window for efficiency. As a result, information from earlier frames may be lost in dynamic scenes with fast motion or extended occlusions. Third, while orthographic projection enables robustness to uncalibrated inputs, it may introduce camera model misalignment in close-range scenes with strong perspective effects. Although our experiments show that the deformation field can largely compensate for these effects, residual distortions can remain; incorporating lightweight intrinsic estimation or perspective-aware refinement is a promising direction. Future work may involve camera estimation model to mitigate camera misalignment and explore efficient mechanisms for adaptively selecting and fusing Gaussians across extended frame histories, which could help retain more temporal context in challenging scenarios.

## F    LICENSES

**Datasets.**

- CO3Dv2 (Reizenstein et al., 2021): CC BY-NC 4.0
- RealEstate10K (Zhou et al., 2018): CC BY 4.0
- DAVIS (Pont-Tuset et al., 2017): BSD 3-Clause License
- YouTube-VOS (Xu et al., 2018): CC BY 4.0

**Pre-trained models.**

- DepthAnythingv2 (Yang et al., 2024b): Apache-2.0 License and CC BY-NC 4.0
- DINOv2 (Oquab et al., 2023): Apache-2.0 License and CC BY 4.0

Table 9: Detailed model configuration of **StreamSplat**.

| Parameter | Value |
|---|---|
| **Image Tokenizer** | |
| **input resolution** | $288 \times 512$ |
| **input channels** | RGB (3) + Depth (1) |
| **patch size** | 8 |
| **output channels** | 768 |
| **Static Encoder** | |
| **layers** | 10 |
| **embed dim** | 768 |
| **attention heads** | 12 |
| **droppath rate** | 0.0 |
| **Dynamic Decoder** | |
| **layers** | 10 |
| **embed dim** | 768 |
| **attention heads** | 12 |
| **droppath rate** | 0.1 |
| **Upsampler** | |
| **layers** | 2 |
| **window size** | 2304 |
| **upsample ratio** | $16\times$ |
| **embed dim** | $768 \rightarrow 192$ |
| **Prediction Heads** | |
| **static head** | Linear |
| **deformation head** | 2-layer MLP |
| **Training** | |
| **optimizer** | AdamW |
| **Adam** $(\beta_1, \beta_2)$ | $(0.9, 0.95)$ |
| **lr scheduler** | CosineAnnealingLR |
| **epochs** | 50 / 70 |
| **batch size** | 128 / 256 |
| **peak learning rate** | $5 \times 10^{-4}$ / $1 \times 10^{-4}$ |
| **weight decay** | 0.05 |
| **gradient clipping** | 1.0 |
| **warm-up iterations** | 20K / 100K |
| **mixed precision** | bf16 |
| **Loss Weights** | |
| $\lambda_{\textbf{MSE}}$ | 1.0 |
| $\lambda_{\textbf{LPIPS}}$ | 0.05 |
| $\lambda_{\textbf{Depth}}$ | 0.05 |
| $\lambda_{\textbf{Mask}}$ | 3.0 |

