# OpenReview forum: "StreamSplat: Towards Online Dynamic 3D Reconstruction from Uncalibrated Video Streams"
_ICLR.cc/2026/Conference — ICLR 2026 Poster_

### Official Review · Reviewer_a3WQ · 2025-10-27

**Soundness:** 3
**Presentation:** 4
**Contribution:** 3
**Rating:** 8
**Confidence:** 3

**Summary:**

This paper proposes **StreamSplat**, a **feed-forward framework for online dynamic 3D reconstruction from uncalibrated video streams**.
It aims to replace conventional optimization-based dynamic 3DGS pipelines with a real-time, fully feed-forward alternative.

**Key contributions include:**
1. **First feed-forward dynamic 3D reconstruction framework** under an *uncalibrated* camera setting.
2. **Effective technical components:**
   - *Probabilistic position sampling* for robust Gaussian position inference under geometric uncertainty.
   - *Bidirectional deformation fields* that jointly model forward and backward motion to ensure temporal coherence.
3. **High-quality results:** Both quantitative tables and qualitative visualizations show consistent improvements and strong temporal coherence across datasets.

**Strengths:**

- **Impressive empirical results:** The method achieves substantial gains over prior 3DGS and NeRF-based approaches, setting a new benchmark for *online dynamic reconstruction from uncalibrated video streams*.
- **Reasonable and clear pipeline:** The proposed two-stage static/dynamic training scheme is well-motivated and reproducible. The combination of a strong image encoder and a dynamic decoder makes architectural sense.
- **Excellent efficiency:** StreamSplat attains orders-of-magnitude speedup (seconds vs. hours) compared to optimization-based methods.
- **Comprehensive evaluation:** The paper compares against strong baselines on multiple datasets, including both static (RE10K, CO3Dv2) and dynamic (DAVIS, YouTube-VOS) settings.

**Weaknesses:**

Major Weaknesses

**W1.** The formulation in *line 157* appears problematic: $(u,v)$ represents pixel coordinates while the offset $o_i$ is in unit space. Their direct addition may be incorrect if the coordinate system is rectilinear. A clarification of this coordinate transformation is needed.

**W2.** Algorithm 2’s *aggregation and fusion* step is ambiguous. The operation `UPDATE` in line 228 is not clearly defined—does it simply replace $\tilde{\mathcal{G}}$ with$ \mathcal{G}_{k-1}^+ $, or else?
Additionally:
- The notation $Î_{t_{k−1}→t_k}$ is unclear—what is the meaning of the arrow?
- The definition of $π$ (projection parameters) is missing.
- The term **“cached”** in line 219 feels implementation-specific and should be formalized or rephrased.
Finally, “rendered frames” in line 221 conflicts conceptually with “dynamic scenes” in line 292—please clarify the intended distinction.

Minor Weaknesses

**W3.** The frame indices in *Figure 2* appear inconsistent. Section 3.2 (line 209) describes deformation between $t=0$ and $t=1$; thus, the figure should use either $(t_n, t_{n-1})$ or $(t_0, t_1)$ for consistency.

**Questions:**

**Q1.** Could you provide a visualization of *multi-point tracking*? Figures 3 and the supplementary videos only illustrate a single tracked point. From my understanding, a per-pixel feed-forward network may struggle with precise point correspondence. Showing multiple tracked points (e.g., ~30) in a local neighborhood would give a fairer sense of temporal coherence.

**Q2.** Please include a **runtime breakdown**—how much time is spent in the encoder, deformation module, and rasterization, respectively? This would help assess real-time feasibility.

---

> ### Author Response · Authors · 2025-11-21
> **Response (1/2)**
>
> Thank you for the positive and insightful comments! In the following, we provide our individual responses and look forward to the subsequent discussion.
>
> > **W1:** The formulation in line 157 appears problematic: $(u, v)$ represents pixel coordinates while the offset $o_i$ is in unit space. Their direct addition may be incorrect if the coordinate system is rectilinear. A clarification of this coordinate transformation is needed.
>
> **A1:** We thank the reviewer for pointing it out. We clarify that this addition is geometrically valid due to our specific **canonical orthographic projection** coordinate.
> + **Our canonical coordinate is not rectilinear.** Unlike methods like pixelSplat which operate in perspective space (where rays diverge from an optical center), StreamSplat operates in a canonical orthographic space to handle uncalibrated inputs from diverse camera. In this canonical space, camera rays are paralleled to the coordinate axes. Therefore, the 3D coordinates $(x, y)$ correspond directly to the normalized pixel coordinates $(u, v)$ adjusted by the learned offset $(o_i)$. Specifically, $u$ and $v$ are normalized device coordinates in $[-1, 1]$, and $o_i$ is a local offset in the same normalized space. Their direct addition yields the correct final 3D position in the canonical volume. We have clarified this coordinate definition in Section 3.1 and Appendix A.
>
> > **W2(a):** Algorithm 2’s aggregation and fusion step is ambiguous. The operation UPDATE in line 228 is not clearly defined—does it simply replace $\tilde{\mathcal{G}}$ with $\mathcal{G}^k_{k-1}$, or else?
>
> **A2(a):** We thank the reviewer for pointing out these notational ambiguities. In L289-290, we explain the UPDATE operation is to "update $\tilde{\mathcal{G}}(t)$ with $\mathcal{G}_{k-1}^{+}(t)$, setting the active deformation of matched 3DGS primitives to the new forward field". We now revise the text to highlight its correlation in Algorithm 2 (mark in blue color).
>
> > **W2(b):** The notation $\hat{I}_ {t_ {k-1} \to t_k}$ is unclear—what is the meaning of the arrow?
>
> > **W2(e):** Finally, “rendered frames” in line 221 conflicts conceptually with “dynamic scenes” in line 292—please clarify the intended distinction.
>
> **A2(b,e):** $\hat{I}_ {t_ {k-1} \to t_k}$ refers to pixel-space RGB output generated by rasterizing that representation, between $t_{k-1}$ and $t_k$, w.r.t. how many t picked between $t_{k-1}$ and $t_k$. We will change the word in L292 to avoid ambiguity.
>
> > **W2(c):** The definition of $\pi$ (projection parameters) is missing.
>
> **A2(c):** We apologize for the missing definition and will clarify it in the revision.
> + **Default Reconstruction:** In our standard uncalibrated setting, $\pi$ defines the fixed canonical orthographic projection (essentially identity rotation/translation with fixed scaling factors $(f_x, f_y)$ matching the patch numbers).
> + **Novel View Synthesis:** When performing novel view synthesis, $\pi$ is updated to represent the target relative camera pose and intrinsic settings required to render the scene from the desired viewpoint.
>
> > **W2(d):** The term “cached” in line 219 feels implementation-specific and should be formalized or rephrased.
>
> **A2(d):**
> + In the **Online Inference Pipeline** (L293), "cached" simply means that we **store** the current Gaussian field $\mathcal{G}_t$ as the persistent scene state and reuse it as input for the next time step $t+1$. There is no special implementation.
> + In the revision, we will replace "cached" with "stored" and explicitly say that we "store the current Gaussian set as the scene state for the next online step." to avoid confusion.
>
> > **W3:** The frame indices in Figure 2 appear inconsistent. Section 3.2 (line 209) describes deformation between $t=0$ and $t=1$; thus, the figure should use either $(t_n, t_{n+1})$ or $(t_0, t_1)$ for consistency.
>
> **A3:** We thank the reviewer for pointing out this notation confusion. In Figure 2, $t_1, t_2, \dots, t_n$ are intended to denote intermediate timestamps sampled within the normalized interval $[0,1]$ between two key frames, with the boundary conditions $t_1 = 0$ and $t_n = 1$. Thus, the deformation in Section 3.2 indeed happens between the endpoints $t=0$ and $t=1$, while the figure visualizes the intermediate times along this interval.
>
> To avoid confusion, we revise the figure and caption to explicitly state that $t_1=0$, $t_n=1$, and that $t_2,\dots,t_{n-1}$ are intermediate timestamps between the two key frames, aligning the notation more clearly with the (t=0, t=1) description in the main text.

---

> ### Author Response · Authors · 2025-11-21
> **Response (2/2)**
>
> > **Q1:** Could you provide a visualization of multi-point tracking? Figures 3 and the supplementary videos only illustrate a single tracked point. From my understanding, a per-pixel feed-forward network may struggle with precise point correspondence. Showing multiple tracked points (e.g., ~30) in a local neighborhood would give a fairer sense of temporal coherence.
>
> **A4:** We appreciate this suggestion and have added multi-point tracking visualizations.
> + In the revised **Appendix C Figure 15**, we now show results where we sample ~30 Gaussians in a local neighborhood and visualize their trajectories jointly over time.
> + We also update the **supplementary video** to include these multi-point tracks, which more clearly demonstrate that our per-pixel feed-forward network produces coherent local trajectories, not just isolated single-point tracks.
>
> > **Q2:** Please include a runtime breakdown—how much time is spent in the encoder, deformation module, and rasterization, respectively? This would help assess real-time feasibility.
>
> **A5:** We thank the reviewer for this suggestion and have added a runtime breakdown in the revised **Appendix C Table 8**.
> On a single A100, the end-to-end runtime per frame is about 49 ms, with approximately:
> + Static Encoder: 18.8 ms (38.4%)
> + Dynamic Decoder: 18.4 ms (37.6%)
> + Rasterization: 4 ms (8.1%)
> + Pre-/post-processing: 7.8 ms (15.9%)

---

> > ### Comment · Reviewer_a3WQ · 2025-11-27
> >
> > Hi, thank you for your detailed responses. You addressed most of my concerns. Now the technical details in the workflow are much clearer to me. And the results of the additional experiments speak: the multi-point tracking performance is intuitively good, and the inference process is fast enough for real-time applications.
> >
> > As for me, the proposed pipeline is clear, and the experiments are sufficient to demonstrate the methodology's validity. So I maintain my judgment that this is a good paper. have a great day.

---

> > > ### Author Response · Authors · 2025-11-28
> > >
> > > Thank you very much for the thoughtful follow-up and for taking the time to review our revisions. We're glad our clarifications addressed your concerns, and we sincerely appreciate your recognition and helpful feedback, which has improved the clarity and completeness of our paper.

---

### Official Review · Reviewer_Soee · 2025-10-31

**Soundness:** 3
**Presentation:** 3
**Contribution:** 3
**Rating:** 8
**Confidence:** 2

**Summary:**

The paper first points out the challenge of achieving real-time dynamic 3D reconstruction from uncalibrated video streams while maintaining the quality and functionality of offline methods. To address this, the paper proposes an online dynamic 3D reconstruction framework called StreamSplat, which integrates probabilistic 3D Gaussian encoding, a bidirectional deformation field, and adaptive Gaussian fusion for temporally coherent and efficient scene modeling. The method aims to enable scalable, feed-forward, and fully online reconstruction of arbitrarily long video streams with state-of-the-art quality and significant speed improvement over optimization-based approaches.

**Strengths:**

* The paper tackles an underexplored yet practically important problem: real-time dynamic 3D reconstruction from uncalibrated video streams, which existing 3DGS and NeRF-based methods generally overlook due to their offline and per-scene optimization nature.

* Unlike prior optimization-based dynamic 3DGS methods, StreamSplat introduces a fully feed-forward pipeline that supports online inference without requiring camera calibration or pre-computed poses, making it highly suitable for real-world deployment scenarios.

* The experiments cover static and dynamic benchmarks, interpolation tasks, and zero-shot evaluation, providing convincing evidence of the model’s robustness, scalability, and temporal coherence.

**Weaknesses:**

* It would be helpful if the authors could clarify whether their framework is capable of predicting or estimating camera poses, given that StreamSplat operates under uncalibrated input conditions. If not, discussing potential extensions in this direction would strengthen the paper's completeness.

* The paper would benefit from additional discussion or experiments on highly dynamic scenes with significant topological changes (e.g., frequent object entries and exits from the field of view). It remains unclear how robust the proposed method is under such conditions.

**Questions:**

See weaknesses.

---

> ### Author Response · Authors · 2025-11-21
>
> Thank you for the positive and insightful comments! In the following, we provide our point-by-point response and look forward to the subsequent discussion.
>
> > **W1:** It would be helpful if the authors could clarify whether their framework is capable of predicting or estimating camera poses, given that StreamSplat operates under uncalibrated input conditions. If not, discussing potential extensions in this direction would strengthen the paper's completeness.
>
> **A1:** StreamSplat does not explicitly predict metric camera poses. Instead, we adopt a fixed canonical space formulation.
> + **Implicit Pose via Deformation:** In our uncalibrated setting, the "camera pose" is effectively absorbed into the deformation fields. The model predicts a deformation that transforms the canonical 3D Gaussians to align with the current observation. This allows us to handle complex camera trajectories (and even non-rigid camera distortions like rolling shutter). This avoids relying on potentially unstable pose estimation in monocular dynamic videos and makes the method applicable even when calibration is unavailable or unreliable.
> + **Potential Extensions:** We agree that explicit pose estimation is a valuable direction. A potential extension would be to add a parallel "Pose Head" that registers the canonical Gaussians to a global coordinate frame. This would allow for applications like AR insertion where a distinct camera trajectory is required. We have discuss this as a valuable future direction in **Appendix E**.
>
> > **W2:** The paper would benefit from additional discussion or experiments on highly dynamic scenes with significant topological changes (e.g., frequent object entries and exits from the field of view). It remains unclear how robust the proposed method is under such conditions.
>
> **A2:** We thank the reviewer for raising this important point, and we want to clarify that our bidirectional deformation and adaptive fusion is designed exactly for this case.
> + **Bidirectional deformation for emerging/disappearing content.** Forward deformation propagates existing Gaussians, while **backward deformation introduces new Gaussians from the newly observed frame** (entries, disocclusions). Adaptive Gaussian fusion then blends them by opacity, fading out outdated Gaussians and fading in newly observed ones, which directly targeting object entries, exits, and occlusions in highly dynamic scenes.
>
> + **Our evaluation on DAVIS with large frame gaps already yields highly dynamic, topology-changing scenes.** By extending DAVIS evaluation to larger temporal gaps, we obtain challenging sequences with rapid motion and topology change. For example, **Figure 5 (Drift Car)** shows a car entering and drifting quickly, and **Figure 12 (Surfing)** shows waves that crash and dissipate, continuously changing surface topology.
>
> + **Further evidence from DyCheck and added qualitative results.** On DyCheck evaluation in Appendix C, we evaluate under large view changes (novel views), stressing our deformation and fusion under strong appearance/visibility changes. **In the revised manuscript L1072-1079 and Figure 16, we also add more qualitative examples with frequent human entries and heavy occlusions**, which further illustrating robustness in highly dynamic real-world scenes.

---

### Official Review · Reviewer_YF1b · 2025-11-01

**Soundness:** 2
**Presentation:** 3
**Contribution:** 3
**Rating:** 4
**Confidence:** 4

**Summary:**

StreamSplat introduced a feed-forward framework for reconstructing dynamic 3D scenes from uncalibrated video streams.

Streamsplat work simliarly as current Dust3R style model and predict the 3D gaussian parameters with a vision transformer.

StreamSplat also introduce three key technical innovation: 1. a probabilistic sampling mechanism 2. a bidirectional deformation field 3. an adaptive Gaussian fusion operation

**Strengths:**

Strength:
1. The paper is well-written and easy to follow.
2. StreamSplat is feedforward and increase the speed of reconstruction.
3. In the figure 4, StreamSpalt shows persistent gaussians across frames, which shows the potential of long-term modeling.

**Weaknesses:**

Major Weakness:

1. Lack of Rigorous Evaluation Protocols: The evaluation does not follow established dynamic reconstruction benchmarks such as DyCheck or NVIDIA Dynamic Scene Dataset. The chosen datasets (DAVIS and YouTube-VOS) are more typical for video segmentation or interpolation, not 4D reconstruction.

2. Limited Training Dataset: The paper uses a mix of static (CO3Dv2, RealEstate10K) and limited dynamic (DAVIS, YouTube-VOS) datasets for training. However, DAVIS contains only a few short clips, this raises concerns about the model’s generalization to complex  motion and real-world application.

3. Dynamic reconstruction typically involves multi-view or multi-camera setups to test geometric consistency (e.g., DyCheck). StreamSplat, in contrast, evaluates on DAVIS using middle-frame interpolation and trivializes the problem.

Minor Weakness:

1.In Figure 1, “curent” should be corrected to “current.”

**Questions:**

Please see the major weakness above.

1. In the section on the Bidirectional Deformation Field, the authors describe how forward and backward deformations enhance temporal consistency. However, given that StreamSplat is evaluated on 5-frame and 8-frame interval tasks, it’s unclear why the model does not increase the number of active Gaussians to better represent accumulated motion over longer intervals.

2. Since the input frames are two, why not use the pretrain model from Croco (e.g. Dust3R, Cut3R)

---

> ### Author Response · Authors · 2025-11-21
> **Response (1/3)**
>
> Thank you for the insightful and detailed comments! Below we briefly summarize our responses to your main concerns:
> +   **Benchmarking:** We provided zero-shot evaluations on **DyCheck and NVIDIA Dynamic Scenes**, where StreamSplat outperforms uncalibrated baselines and approaches calibrated methods.
> +   **Dataset & protocol:** We clarified that **DAVIS and YouTube-VOS** are standard, essential benchmarks for evaluating complex non-rigid motion and temporal consistency for dynamic reconstruction methods.
> +  **Design:** We clarify how bidirectional deformation and adaptive fusion control the number of active Gaussians over long intervals, and why we do not rely on Croco-style pretrained models.
>
> We hope this summary clarifies our overall position. In the following, we provide detailed point-by-point responses and look forward to the subsequent discussion.
>
> > **W1(a):** Lack of Rigorous Evaluation Protocols: The evaluation does not follow established dynamic reconstruction benchmarks such as DyCheck or NVIDIA Dynamic Scene Dataset.
>
> **A1(a):**
> We thank the reviewer for this comment.
>
> **1. Evaluations on Existing Conventional Benchmarks**
> + We respectfully clarify that we **already included** these evaluations in the submission, and we have expanded them in this rebuttal.
>     + **Zero-Shot DyCheck evaluation is already included (Appendix C).** We evaluated StreamSplat on DyCheck under an uncalibrated protocol following DGMarbles in the original submission (**Table 6, Figure 14**). StreamSplat (w/o cam) achieves 12.37 dB PSNR, significantly outperforming the uncalibrated baseline DGMarbles (w/o cam) (9.76 dB) and matching calibrated 4DGS (12.75 dB) which relies on ground-truth intrinsics
>     + **Additional zero-Shot on NVIDIA Dynamic Scenes.** Per your suggestion, we added a **zero-shot** evaluation on the NVIDIA Dynamic Scene dataset. To ensure a fair comparison against uncalibrated methods, we evaluated DGMarbles using intrinsics estimated by CUT3R (as mentioned in your Q2), while providing it with GT poses and scene metadata (scale, near/far) to ensure valid execution. StreamSplat (w/o camera) significantly outperforms DGMarbles (w/ CUT3R) and approaches the performance of 4DGS (w/ GT camera). We list the average results below (full details in revised **Appendix C, Table 7 and revised Figure 14**):
>
> |        Method       | Extrinsic | Intrinsic | Scene Info. |  PSNR / LPIPS |
> |:-------------------:|:---------:|:---------:|:-----------------:|:-------------:|
> |    4DGS (w/ cam)    |     GT    |     GT    |    $\checkmark$   | 17.59 / 0.292 |
> | DGMarbles (w/ CUT3R) |     GT    |   CUT3R   |    $\checkmark$   | 13.91 / 0.595 |
> |         **StreamSplat (Ours)**        |  Relative |  $\times$ |      $\times$     | **16.30 / 0.486** |
>
> **2. Discussion on Metric Alignments**
> + While we report these results for comparison, we note that the **standard protocols are fundamentally ill-posed for uncalibrated methods**, as they enforce a specific metric world that is **unknowable** in our setting. **Nonetheless, StreamSplat still outperforms uncalibrated baselines under these protocols, even though they rely on additional scene information that we do not use.**
>     + **Scale ambiguity caused by missing calibrated camera prior.** Conventional benchmarks assume a unique, metric 3D world defined by exact calibration (intrinsics/extrinsics) and global scale, using a perspective projection model. Our method, however, reconstructs in an orthographic canonical space but not anchored to a specific metric camera model. Consequently, **the 3D scene cannot be uniquely determined**; valid reconstructions exist only in a "canonical" space, which may be arbitrarily scaled or shifted relative to the benchmark ground truth.
>     + **Our goal is robust generalization to diverse real-world inputs,** not specialization to a single benchmark coordinate system. StreamSplat is designed to handle diverse in-the-wild videos (e.g., fisheye and pinhole cameras). By absorbing varying camera parameters and depth errors into the Gaussian dynamics, we generate accurate reconstructions in a flexible canonical space. This naturally introduces metric misalignment with fixed benchmark coordinates, but it does not compromise the underlying reconstruction quality.

---

> ### Author Response · Authors · 2025-11-21
> **Response (2/3)**
>
> > **W1(b):** The chosen datasets (DAVIS and YouTube-VOS) are not for 4D reconstruction.
>
> **A1(b):** We would like to clarify that DAVIS and YouTube-VOS are indeed suitable for 4D reconstruction. They are standard benchmarks consisting of challenging in-the-wild monocular videos, unlike controlled studio-captured 4D datasets.
>
> + **Evaluating on these datasets is standard practice for recent dynamic reconstruction works [1,2].** SOTA dynamic methods such as Splatter-A-Video [1] and MonST3R [2] adopt DAVIS-style videos to evaluate dynamic 3D behavior and temporal consistency under monocular, uncalibrated settings. We follow this established practice to ensure direct comparisons with prior art.
>
> + **These datasets feature in-the-wild scenes with complex, non-rigid motion that dynamic reconstruction should handle.** Unlike mostly rigid SfM benchmarks, DAVIS and YouTube-VOS contain diverse foreground objects with large non-rigid motion, significant deformations, frequent occlusions, and sudden changes (objects entering/leaving). These are critical challenges for modeling practical dynamic 3D fields in the wild, which traditional benchmarks often lack.
>
> > **W3:** Dynamic reconstruction typically involves multi-view or multi-camera setups to test geometric consistency (e.g., DyCheck). StreamSplat, in contrast, evaluates on DAVIS using middle-frame interpolation and trivializes the problem.
>
> **A3:** We respectfully disagree that using DAVIS trivializes the problem; **it still rigorously evaluates geometric consistency through temporal evolution**.
>
> + **Our evaluation enforces 3D geometric consistency under camera ego-motion and complex object dynamics.** Following and extending recent dynamic protocols (Splatter-A-Video [1], MonST3R [2]), we use a challenging sparse setting with 5–8 frame gaps on DAVIS/YouTube-VOS. In this uncalibrated monocular setting, the model must reconstruct a continuous and coherent 4D representation:
>     + **Camera Motion (Parallax):**  Large ego-motion induces strong parallax, so the model must predict accurate geometry to render from new time and viewpoints.
>     + **Object Motion (Topology):** Rapid object motion introduces complex surface deformations and topological changes (self-occlusions/disocclusions), where a trivial 2D interpolation cannot maintain structural consistency (see Table 2).
>
> As the reviewer kindly noted, our results show "persistent Gaussians across frames", and in the revision we further visualize multiple Gaussians in a local neighborhood (please refer to **a3WQ's A4**, Figure 15). This persistence under both camera parallax and object deformation directly indicates a stable, coherent 4D reconstruction rather than frame-wise image processing.
>
> + Conventional multi-view benchmarks are also utilized as complementary validation. We provide **zero-shot results on DyCheck and NVIDIA Dynamic Scenes**, where StreamSplat consistently outperforms uncalibrated baselines and approaches calibrated ones; please refer to **A1(a)** for details.
>
> > **W2:** Limited Training Dataset: The paper uses a mix of static (CO3Dv2, RealEstate10K) and limited dynamic (DAVIS, YouTube-VOS) datasets for training. However, DAVIS contains only a few short clips, this raises concerns about the model’s generalization.
>
> **A2:**
> We agree that our training data might be limited regarding the coverage of various dynamic scenarios, **but relative to existing dynamic reconstruction methods, StreamSplat already demonstrates substantially stronger generalization.**
>
> + **StreamSplat is a general model with demonstrated zero-shot generalization.** Unlike baselines that require per-scene optimization (e.g., DGMarbles, 4DGS) or are per-dataset training (e.g., pixelSplat), StreamSplat is trained once as a generalist feed-forward model on mixed static + dynamic data and then applied zero-shot to unseen videos. As shown in **A1(a)**, it achieves strong zero-shot performance on DyCheck and NVIDIA, indicating that it learns transferable 3D motion priors rather than overfitting to a few training clips.
>
> + **Dynamic supervision is not limited to a few DAVIS clips.** While DAVIS (90 high-resolution videos) is relatively small, our dynamic training comes mainly from **YouTube-VOS (4k+ videos)**, with DAVIS used as an additional high-quality source. Combined with static scenes from CO3Dv2 and RealEstate10K, this strategic mix exposes the model to a wide range of geometric structures and dynamic motions.
>
> + **Scaling dynamic data is a natural next step.** We agree that training on larger, more diverse datasets would likely further improve generalization. Our current experiments are constrained by our computational resources, but our feed-forward design is inherently efficient and scalable, making larger-scale training a natural direction for future work. **We hope the reviewer agree that, within a reasonable computational and data budget, our StreamSplat has already demonstrated strong generalization.**

---

> ### Author Response · Authors · 2025-11-21
> **Response (3/3)**
>
> > **W4:** In Figure 1, “curent” should be corrected to “current.”
>
> **A4:** Thank you for pointing it out. We already revised in the revised pdf.
>
> > **Q1:** In the section on the Bidirectional Deformation Field, the authors describe how forward and backward deformations enhance temporal consistency. However, given that StreamSplat is evaluated on 5-frame and 8-frame interval tasks, it’s unclear why the model does not increase the number of active Gaussians to better represent accumulated motion over longer intervals.
>
> **A5:**
> StreamSplat adopts **one-time explicit expansion** with **automatic adaptation** of the number of active Gaussians.
>
> + **Our framework is observation-grounded reconstruction and does not introduce more "virtual" Gaussians just because the interval is longer.**  At the beginning of bidirectional fusion, we explicitly **double** the candidate set by taking the union of Gaussians from the forward- and backward-deformed fields. Importantly, both come from **observed** key frames, so geometry and texture remain tied to real pixels. We do not introduce additional "virtual" Gaussians solely because the temporal interval is longer, which would risk hallucinated structure and is undesirable for a reconstruction method.
>
> + **Adaptive fusion automatically adjust how many Gaussians remain active for long intervals.** We explicitly handle new content via the Backward Deformation Field, which adds Gaussians from the newly observed frame. Our adaptive Gaussian fusion then performs soft matching between forward- and backward-warped Gaussians: when motion is small, many Gaussians match and merge; for larger 5–8 frame gaps, fewer matches occur, so more Gaussians stay independently "active". In this way, the **effective number of active Gaussians is automatically adjusted according to motion magnitude and scene dynamics**, rather than manually specified.
>
> This adaptive strategy removes the need for iterative densification or heuristic pruning in optimization-based methods, and is a core innovation that allows StreamSplat to process arbitrarily long videos in a fully feed-forward, online manner while maintaining robust temporal coherence.
>
>
> > **Q2:** Since the input frames are two, why not use the pretrain model from Croco (e.g. Dust3R, Cut3R)
>
> **A6:** While CroCo/Dust3R-style models are SOTA for static geometry and pose estimation, they are fundamentally misaligned with our goal of online dynamic 3DGS reconstruction for three reasons:
>
> + **Representation Mismatch (dense static points vs. dynamic Gaussians):** Dust3R-style models predict very dense scene-coordinate point maps, which are hard to deform efficiently for continuous dynamics. In 3DGS, SfM points are only a rough initialization; effective Gaussians must move and change opacity to represent volume. Directly using Dust3R points as Gaussians would be both computationally prohibitive and still not yield a compact, deformable 4D Gaussian field.
>
> + **Probabilistic Sampling vs. Deterministic Regression:** Dust3R-style models performs deterministic regression of 3D points, whereas our probabilistic sampling head is crucial for feed-forward 3DGS: as shown in Table 4 (and also observed in pixelSplat), purely deterministic prediction leads to local minima and blurry reconstructions. Dust3R's pretrained weights do not support our sampling-based Gaussian prediction.
>
> + **Static pose priors vs. dynamic, uncalibrated streams:** Dust3R-style models is trained to find static matches and explicit camera poses. In monocular dynamic videos, such pose estimation can be unreliable and can even harm dynamic modeling (see our DGMarbles (w/ CUT3R) comparison in **A1(a)**). We therefore deliberately avoid explicit camera estimation and model all motion via our dynamic decoder.
>
> Given these mismatches, we find Dust3R-style features and poses are not directly suitable for dynamic 3DGS reconstruction, and thus we didn't use their pretrained models.
>
>
> [1] Sun, Y.T., et al. Splatter a video: Video gaussian representation for versatile processing. NeurIPS 2024.
>
> [2] Zhang, J., et al. MonST3R: A Simple Approach for Estimating Geometry in the Presence of Motion. ICLR 2025.

---

> ### Author Response · Authors · 2025-11-25
>
> Dear Reviewer YF1b,
>
> Thank you again for your thoughtful feedback and the time you have spent on our paper. We hope our rebuttal has addressed your concerns.
>
> With the discussion deadline approaching, please do not hesitate to let us know if any questions remain. We would greatly appreciate the opportunity to further discuss them.
>
> Best regards,
>
> Authors

---

### Author Response · Authors · 2025-11-21
**Revision Summary**

Dear Reviewers,

We sincerely thank you for your constructive feedback.

We appreciate the reviewers' recognition that StreamSplat (1) tackles an important and practically relevant problem (**Soee**) while attaining orders-of-magnitude speedup over optimization-based methods (**YF1b**, **a3WQ**),
(2) provides a comprehensive and diverse evaluation with impressive empirical results across static/dynamic benchmarks, and zero-shot evaluations (**Soee**, **a3WQ**),
and (3) achieves temporally coherent reconstruction with persistent Gaussians across frames (**YF1b**).

In response to these comments, we have updated the manuscript by:
+ Adding additional zero-shot evaluations on the NVIDIA Dynamic Scene benchmark (text L1065–1071, Table 7, Figure 14).

+ Adding multi-point tracking visualizations to better illustrate local temporal coherence (text L236, Figure 15, and supplementary video).

+ Providing a detailed runtime breakdown of the encoder, decoder, and rasterization (text L1130–1133, Table 8).

+ Expanding the results and discussion for highly dynamic, topology-changing scenes (text L1072–1079, Figure 16).

+ Clarifying algorithms and figures to avoid confusion, including updates to Figure 1, Figure 2, Algorithm 2, and the associated description of the online pipeline (L285–292).

We hope that these revisions and clarifications will help address your concerns.
We now respond to the individual reviewers below and are happy to discuss more if necessary.

Best regards,

Authors

---

### Author Response · Authors · 2025-12-03
**Post-Rebuttal Summary**

Dear Area Chairs and Reviewers,

We sincerely thank all reviewers for their time and constructive feedback, and all ACs and organizers for upholding high standards of review and research integrity despite this year's unexpected incident. As the discussion phase concludes, we provide a brief summary of the reviewer consensus and how we resolved the remaining concerns to assist in the final assessment.

+ **Reviewer consensus on strengths.** We are encouraged that Reviewers **Soee** and **a3WQ** recommend acceptance, and that all three reviewers recognize the strengths of our paper:
    + **Novelty:** a "novel solution" to an "underexplored yet practically important problem" (uncalibrated online dynamic reconstruction) (**Soee**).
    + **Efficiency:** "orders-of-magnitude speedup" (~49ms/frame vs. hours) and "real-time feasibility" (**a3WQ**, **YF1b**).
    + **Quality:** "comprehensive evaluation" with "impressive empirical results" (**a3WQ**) and temporally coherent reconstruction (**YF1b**).

+ **Resolution of remaining concerns.** During the rebuttal, we addressed key questions and clarified **factual misunderstandings**, particularly regarding evaluation:
    + **"Missing" Benchmarks (YF1b):** We clarify that **zero-shot DyCheck results were already included in the original submission** (Appendix C). We additionally **added zero-shot NVIDIA Dynamic Scenes** during rebuttal, where StreamSplat outperforms uncalibrated baselines and approaches calibrated oracles (Table 7).
    + **Datasets and Generalization (YF1b)**: We clarified that DAVIS is **standard** for evaluating non-rigid motion in recent dynamic reconstruction works, and our dynamic supervision is primarily from **YouTube-VOS (4k+ videos)** rather than DAVIS alone; **generalization** is further supported by our **zero-shot results** on unseen datasets.
    + **Robustness and Tracking** (**Soee**, **a3WQ**): We added **multi-point tracking** visualizations (Figure 15) and expanded results on **topology-changing scenes** (Figure 16). Reviewer **a3WQ** confirmed these results addressed their concerns and maintained their accept recommendation.

With these added evidence and the clarification of factual misunderstandings, we hope all outstanding concerns are resolved.

Best regards,

Authors

---

### Meta-Review · Area_Chair_hEYb · 2026-01-05

**Summary:**

This paper proposes a first feedforward method for dynamic Gaussian reconstruction with uncalibrated videos. All reviewers appreciated the novelty, technical insights and the impressive results of the work.

The reviewers' primary concerns were the following.
1. Results on the DyCheck and NVIDIA datasets.
2. Clarification on whether the method can predict camera poses and visual results on large topological changes.
3. Clarification of the method and additional visual results on tracking multiple points on images.

They were all adequately addressed by the authors' rebuttal.

**Reviewer Concerns:**

Concerns addressed by the rebuttal:
1. Results on the DyCheck and NVIDIA datasets.
2. Clarification on whether the method can predict camera poses and visual results on large topological changes.
3. Clarification of the method and additional visual results on tracking multiple points on images.

Concerns not addressed by the rebuttal:
None.

**Reviewer Scores:**

1. Reviewer YF1b (Rating: 4: marginally below the acceptance threshold. But would not mind if paper is accepted)

The reviewer's primary concern was the lack of evaluations on dynamic reconstruction benchmarks including DyCheck and the NVIDIA Dynamic Video Dataset. The authors provided evaluations on these benchmarks and show their method to be superior to the baselines, numerically. The author's response around the "Discussion on Metric Alignments" to this reviewer is not relevant to the reviewers' concerns. Overall, the reviewer would have likely raised their score.

2. Reviewer Soee (8: accept, good paper (poster))

The reviewer's primary concern was (a) clarifying if the method predicts camera poses or not and (b) clarifying if the method can handle highly dynamic scenes with significant topological changes. The authors adequately addressed both points in the rebuttal. Hence, the reviewer is likely to have maintained their positive rating.

3. Reviewer a3WQ (8: accept, good paper (poster))

Their concerns were primarily around requiring (a) additional clarifications of the method and (b) additional visualizations. All these concerns were adequately addressed. Hence, the reviewer is likely to have maintained their positive rating.

---

### Decision · Program_Chairs · 2026-01-26

Accept (Poster)